# Apolipoprotein L-1 renal risk variants form active channels at the plasma membrane driving cytotoxicity

Joseph A Giovinazzo[1]*, Russell P Thomson[1], Nailya Khalizova[1], Patrick J Zager[2], Nirav Malani[3], Enrique Rodriguez-Boulan[2], Jayne Raper[1]*, Ryan Schreiner[2]*

[1]Department of Biological Sciences, Hunter College at City University of New York, New York, United States; [2]Department of Ophthalmology, Margaret Dyson Vision Research Institute, Weill Cornell Medicine, New York, United States; [3]Genosity, Iselin, United States

**Abstract** Recently evolved alleles of Apolipoprotein L-1 (*APOL1*) provide increased protection against African trypanosome parasites while also significantly increasing the risk of developing kidney disease in humans. APOL1 protects against trypanosome infections by forming ion channels within the parasite, causing lysis. While the correlation to kidney disease is robust, there is little consensus concerning the underlying disease mechanism. We show in human cells that the APOL1 renal risk variants have a population of active channels at the plasma membrane, which results in an influx of both $Na^+$ and $Ca^{2+}$. We propose a model wherein APOL1 channel activity is the upstream event causing cell death, and that the activate-state, plasma membrane-localized channel represents the ideal drug target to combat APOL1-mediated kidney disease.

**\*For correspondence:**
joseph.giovinazzo@cuanschutz.edu (JAG);
raper@genectr.hunter.cuny.edu (JR);
ryanschreiner@gmail.com (RS)

**Competing interests:** The authors declare that no competing interests exist.

## Introduction

Apolipoprotein L-1 (*APOL1*) is a primate-specific innate immunity gene, (*Smith and Malik, 2009*) which provides protection against protozoan parasites (*Hager et al., 1994*; *Samanovic et al., 2009*) by forming cation channels within the pathogens (*Molina-Portela et al., 2005*; *Thomson and Finkelstein, 2015*). APOL1 circulates on specialized high-density lipoprotein particles termed trypanosome lytic factors, which are endocytosed by the parasites (*Hager et al., 1994*; *Rifkin, 1978*; *Raper et al., 1999*). Once inside, APOL1 leads to an ion flux that drives trypanolysis (*Molina-Portela et al., 2005*; *Thomson and Finkelstein, 2015*; *Rifkin, 1984*). Its activity is governed by a two-step process: Activation at acidic pH followed by channel opening at neutral pH (*Thomson and Finkelstein, 2015*). This mechanism is inhibited in human-infective *Trypanosoma brucei rhodesiense* (*Pérez-Morga et al., 2005*) and *T.b. gambiense* (*Capewell et al., 2013*; *Uzureau et al., 2013*), leading to sleeping sickness.

A molecular arms race between humans and African trypanosomes has led to the evolution of African *APOL1* variants, G1 (rs73885319 - S342G, rs60910145 - I384M) and G2 (rs71785313 - Δ 388:389 NY) (*Genovese et al., 2010*), which provide protection against the human infective trypanosomes (*Cooper et al., 2017*). This resistance, however, significantly increases the risk of developing a spectrum of chronic kidney diseases when two copies of these renal risk variants (RRVs) are present, including focal segmental glomerulosclerosis, hypertension-associated end stage kidney disease, and HIV-associated nephropathy (*Genovese et al., 2010*; *Kopp et al., 2011*; *Tzur et al., 2010*). The RRVs are also associated with sickle cell nephropathy (*Ashley-Koch et al., 2011*) and lupus nephritis (*Freedman et al., 2014*), and drive faster progression from chronic kidney disease to renal failure (*Parsa et al., 2013*). Importantly, 5 million African Americans are estimated to carry two copies of G1 or G2 (*Friedman et al., 2011*).

The major isoform of *APOL1* encodes a signal peptide (*Nichols et al., 2015*; *Monajemi et al., 2002*) and likely traffics along the secretory pathway, thereby allowing for secretion from hepatocytes onto high density lipoprotein particles (*Shukha et al., 2017*) or localization to the endoplasmic reticulum (ER) and plasma membrane (PM) in other cell types (*Cheng et al., 2015*; *O'Toole et al., 2018*; *Olabisi et al., 2016*; *Heneghan et al., 2015*). The majority of intracellular APOL1 remains localized within the ER (*Cheng et al., 2015*). *APOL1* is expressed by several kidney cell types including the podocyte (*Nichols et al., 2015*; *Ma et al., 2015*), and multiple studies point to kidney intrinsic APOL1 as the driver of disease (*Reeves-Daniel et al., 2011*; *Lee et al., 2012a*), rather than the circulating APOL1 associated with trypanosome lytic factors (*Kozlitina et al., 2016*). While the discovery of the RRVs provided an explanation for the increased rates of kidney disease in African Americans, there remains little consensus on how the variants cause disease or which pathways to target for therapeutic intervention.

Overexpression of the RRVs in multiple cell lines and transgenic mouse models causes cytotoxicity, however the mechanism responsible remains unclear. It has been proposed that RRV cytotoxicity is mediated by several possible pathways such as autophagy (*Wan et al., 2008*), lysosomal permeability (*Lan et al., 2014*), pyroptosis (*Beckerman et al., 2017*), mitochondrial dysfunction (*Ma et al., 2017*), impairment of vacuolar acidification (*Kruzel-Davila et al., 2017*), activation of stress-activated kinases (*Olabisi et al., 2016*), and ER stress (*Wen et al., 2018*). This lack of consensus is unsatisfactory and hinders progress towards developing therapeutics. However, whilst these pathways are seemingly unrelated, most are affected by or activated to combat pore-forming toxins (*Huffman et al., 2004*; *Cancino-Rodezno et al., 2009*; *Kennedy et al., 2009*). Therefore, as APOL1 forms cation channels within trypanosomes after endocytosis (*Molina-Portela et al., 2005*; *Thomson and Finkelstein, 2015*), we hypothesize cell intrinsic G1 and G2 also form cytotoxic channels, and that this mechanism links the disparate pathways together.

To perform this study, we focused on the channel forming properties of APOL1. Interestingly, APOL1 led to an intracellular accumulation of $Ca^{2+}$ after 72 hr of overexpression in *Xenopus* oocytes (*Heneghan et al., 2015*), and $Ca^{2+}$ signaling has been associated with the activation of several aforementioned pathways linked to APOL1 (*Lee et al., 2012b*; *Rizzuto et al., 2012*; *Krebs et al., 2015*). Additionally, treatment of African trypanosomes with human serum led uptake of $Ca^{2+}$ (*Rifkin, 1984*). The APOL1 channel is permeable to monovalent $Na^+$ and $K^+$ (*Thomson and Finkelstein, 2015*), and its trypanolytic activity is inhibited by reducing extracellular $Na^+$ (*Molina-Portela et al., 2008*). As the plasma membrane is already highly permeable to $K^+$, we focused on the potential roles of extracellular $Na^+$ and $Ca^{2+}$ in driving APOL1 cytotoxicity. We utilized planar lipid bilayers to evaluate APOL1 as a possible non-selective cation channel, and live-cell fluorescent microscopy with the cytoplasmic $Ca^{2+}$ indicator GCaMP6f (*Chen et al., 2013*) and membrane voltage sensor FliCR (*Abdelfattah et al., 2016*) to test for increased $Ca^{2+}$ and $Na^+$ flux linked to RRV-induced cytotoxicity. Furthermore, utilizing the retention using selective hooks (RUSH) system (*Boncompain et al., 2012*), in combination with live-cell and immunofluorescence microscopy, we evaluated the importance and timing of events leading up to RRV-induced cell death, including ER exit and the delivery of APOL1 to the PM.

## Results

### Expression of the *APOL1* renal risk variants G1 and G2, but not G0, leads to cell death

As an innate immunity gene, *APOL1* is induced by pro-inflammatory cytokines such as interferons (*Nichols et al., 2015*). Prolonged courses of interferon treatment caused acute emergence of collapsing focal segmental glomerulosclerosis in a small subset of patients who were revealed to carry two copies of the RRVs upon retrospective genotyping (*Markowitz et al., 2010*). This has led to the hypothesis that a sustained increase in RRV expression is a cause of APOL1-driven chronic kidney disease (*Nichols et al., 2015*; *Olabisi et al., 2016*).

A Flp-In TREX 293 (FT293) stable cell line was generated to inducibly express *APOL1* variants from a single genetic locus, allowing us to test the effects of sustained *APOL1* expression. These variants are based on the most prevalent haplotypes in the human population (*Auton et al., 2015*; *Figure 1a*). Expression of the *APOL1* variants leads to similar levels of protein expression

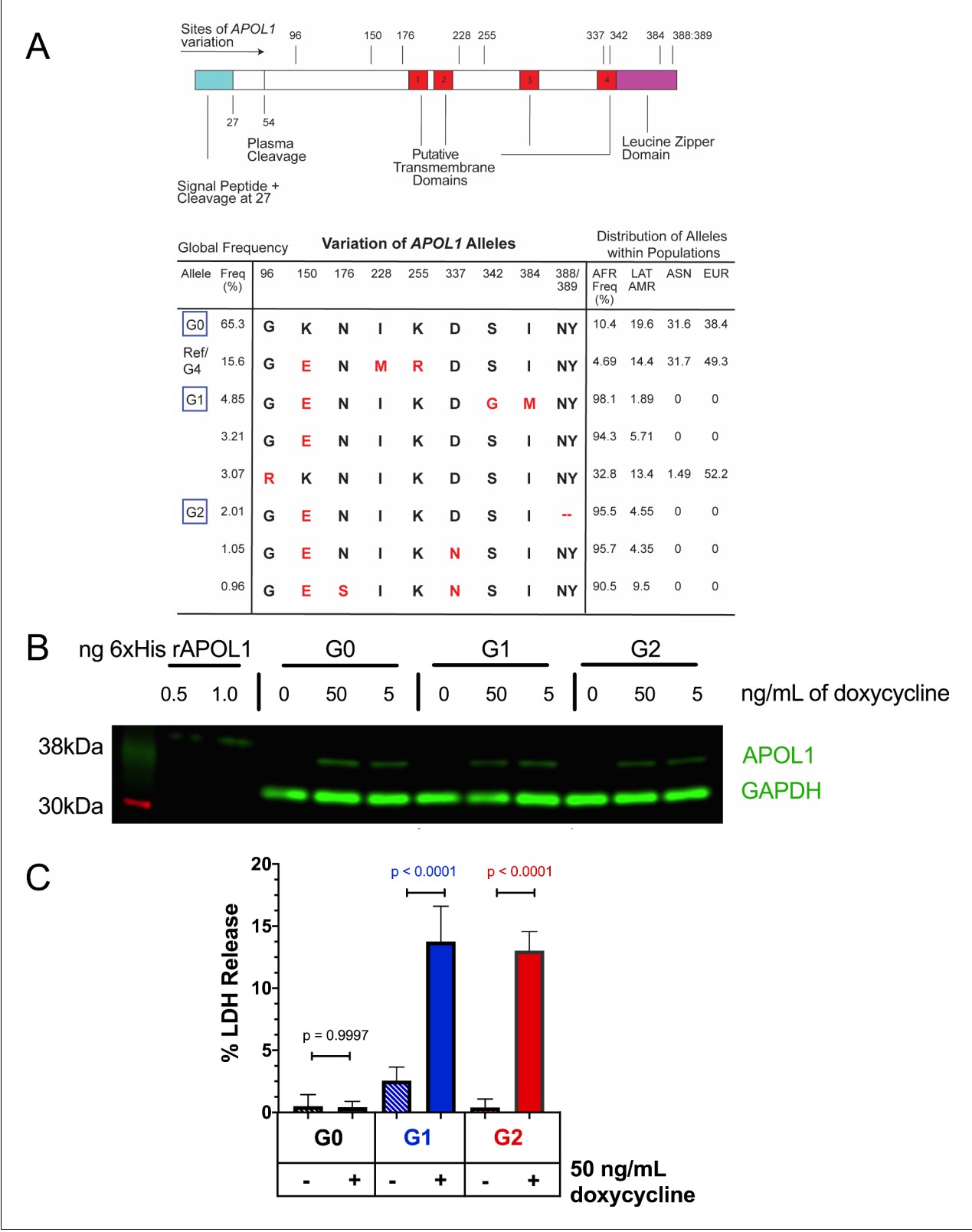

**Figure 1.** Expression of *APOL1-G1* and *G2* cDNA leads to cytotoxicity in FT293 cells. (a) Predicted linear structure of APOL1, using JPred, with major sites of amino acid variation highlighted in red (a deletion is represented as a dash). Haplotypes are organized by frequency in the human population, which is depicted in the left-hand column as Freq (%). The right-hand column represents the distribution of each allele within populations. AFR = African, LAT AMR = Latin America, ASN = Asian, EUR = European. Haplotypes in blue boxes were those used in this study. Data retrieved from

*Figure 1 continued on next page*

Figure 1 continued

1000 Genomes Project. (**b**) Western blot of whole cell lysates displaying similar levels of protein production between FT293 cell lines. Cells were treated with doxycycline for 4 hr. 6x-His tagged APOL1 was expressed and purified from *E. coli* and used as a positive control. (**c**) Cell death assay displaying the cytotoxicity caused by doxycycline-induced expression of *APOL1-G1* and *G2,* but not *G0*, in FT293 cells. Cells were induced with 50 ng/mL doxycycline for 24 hr, and cytotoxicity was measured via cellular release of lactate dehydrogenase. A two-way ANOVA with multiple comparisons was performed to compare induced and un-induced cells (n = 14).

The online version of this article includes the following figure supplement(s) for figure 1:

**Figure supplement 1.** Expressing APOL1 protein at levels found in podocytes leads to RRV cytotoxicity in FT293 cells.

(*Figure 1b*), and induction of the RRVs, but not G0, leads to cell swelling followed by cytotoxicity after 24 hr of expression (*Figure 1c*, *Figure 1—figure supplement 1a*).

It has recently been suggested that overexpression of *APOL1* in cultured cells may not constitute a physiologically relevant model as lower expression levels are not cytotoxic (*O'Toole et al., 2018*). However, no reference for *APOL1* expression has been established for comparison. To address this point, we titrated APOL1 protein expression in the FT293-G0 stable cell line to obtain similar APOL1 levels found in interferon-stimulated human podocytes (*Saleem et al., 2002*). The RRVs remained cytotoxic in our model under these conditions, though cell death was delayed due to lower expression (*Figure 1—figure supplement 1b–c*). These findings indicate that RRV-mediated cytotoxicity in this cell system occurs with levels of protein expression, as quantified by western blot, comparable to that found in interferon-stimulated podocytes, and represents a productive cell culture model to evaluate APOL1-mediated kidney disease.

## APOL1 channels are permeable to Ca²⁺, and the RRVs lead to a cellular Ca²⁺ influx

We hypothesize that the cytotoxicity of APOL1 in mammalian cells parallels its trypanolytic activity, both resulting from its channel-forming properties. Indeed, APOL1 causes cell swelling and dissipation of $Na^+$ and $K^+$ gradients in trypanosomes (*Molina-Portela et al., 2005*; *Rifkin, 1984*) as well as mammalian cells (*O'Toole et al., 2018*; *Olabisi et al., 2016*). Furthermore, overexpression of APOL1 in *Xenopus* oocytes led to an intracellular accumulation of $Ca^{2+}$ (*Heneghan et al., 2015*). As $Ca^{2+}$ is a potent signaling molecule, aberrantly high cytoplasmic $Ca^{2+}$ levels can activate many cell-signaling pathways; eventually its dysregulation leads to cell death.

The potential $Ca^{2+}$-permeability of channels formed by recombinant APOL1 (rAPOL1) was examined using planar lipid bilayers (*Figure 2a*). $CaCl_2$ was first added to both sides of the bilayer in equimolar amounts. Under conditions where $CaCl_2$ was present on both sides of the bilayer, but not KCl or NaCl, the rAPOL1 channel retained its pH-dependent activity, requiring a pH $\leq 6.0$ for irreversible membrane insertion followed by neutralization to open, enhancing conductivity several hundred-fold (*Figure 2b*;

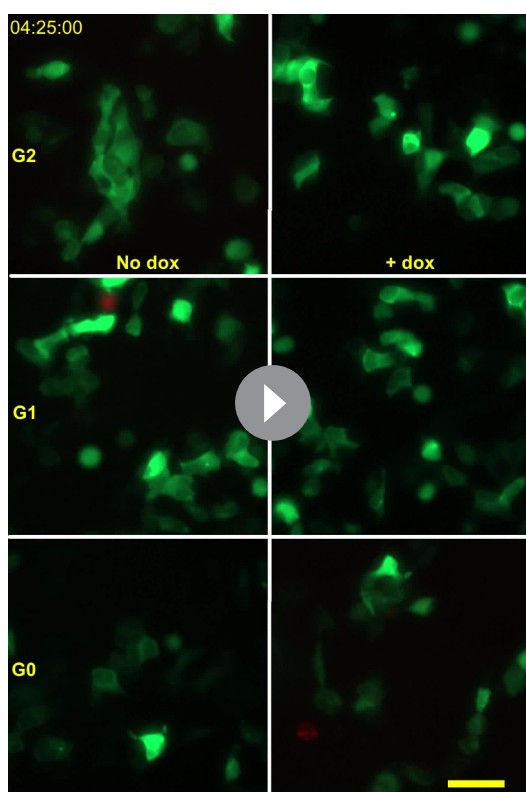

**Video 1.** Expression of G1 and G2 leads to a $Ca^{2+}$ influx prior to cell swelling. FT293 cells were transfected with GCaMP6f 24 hr before imaging. Cells were then incubated with 3 µM DRAQ7 and with or without 50 ng/mL doxycycline to induce *APOL1* expression. Cells were imaged via widefield from 4.5 to 30 hr post induction, and dual color images were taken every 10 min. Scale bars = 50 µm.

https://elifesciences.org/articles/51185#video1

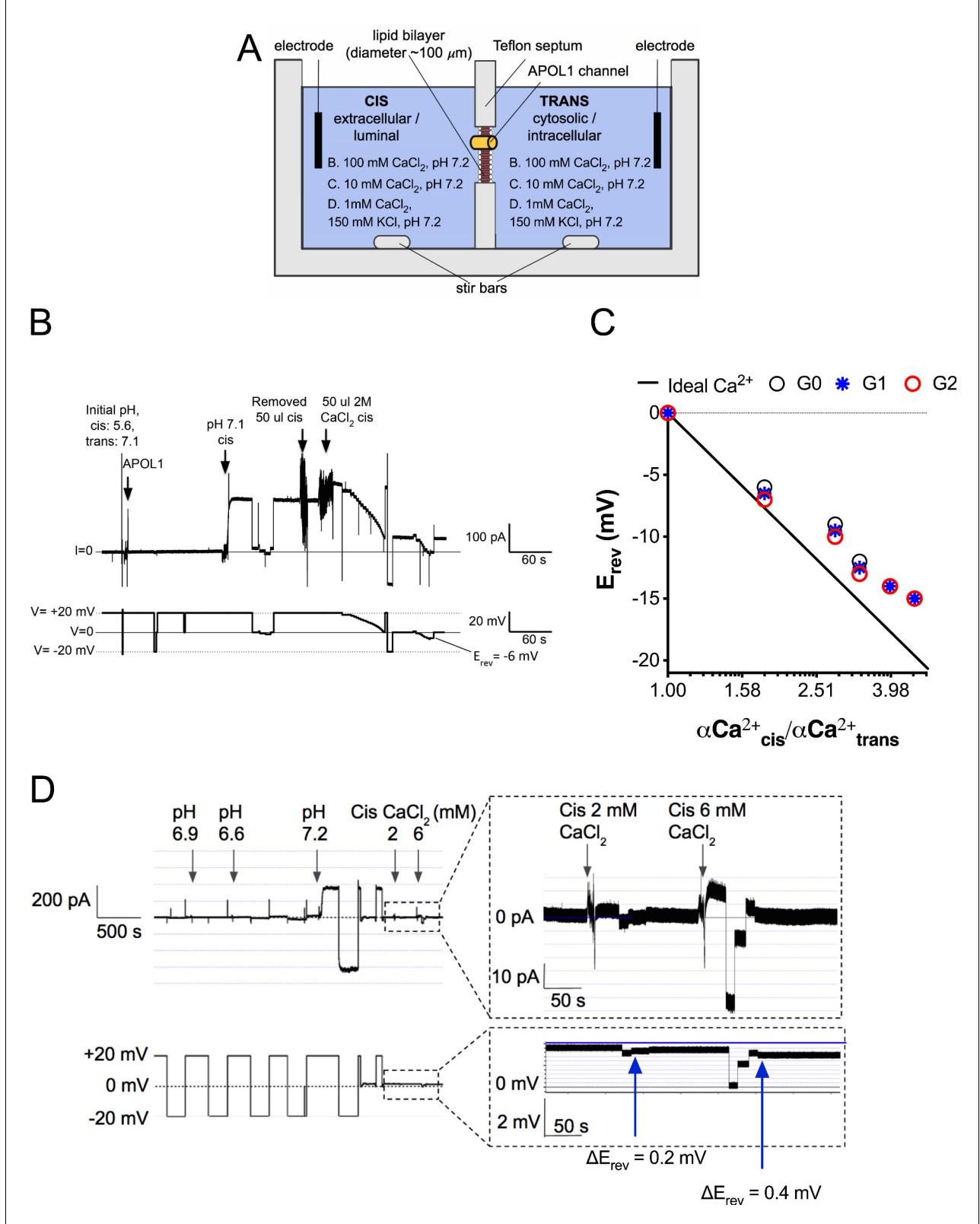

**Figure 2.** The APOL1 channel is permeable to Ca²⁺. (a) Planar lipid bilayer setup. The starting buffer composition for (b–d) are shown. During each experiment the composition of the cis side is altered by the experimenter, whereas the trans side is left unaltered. After APOL1 channel formation (typically many thousands per bilayer) a current (pA, upper trace) can be measured in response to a voltage (V, lower trace). In each case the voltage is set by the experimenter. (b) Planar lipid bilayer demonstrating that the rAPOL1-G0 channel is selective for Ca²⁺ over Cl⁻. rAPOL1 was added to the cis

*Figure 2 continued on next page*

*Figure 2 continued*

side at pH 5.6 to drive insertion, which caused a minor increase in conductance that was amplified approximately 450-fold upon cis neutralization (pH 7.1). The voltage required to zero the current (reversal potential, $E_{rev}$) with a 1.95-fold $CaCl_2$ gradient was −6 mV, indicating $Ca^{2+}$ selectivity. (c) $Ca^{2+}$ versus $Cl^-$ permeability did not differ between APOL1 G0, G1 and G2. A conductance was obtained as in **b**, except that the chambers contained symmetrical 10 mM $CaCl_2$. The $E_{rev}$ was determined as $CaCl_2$ was titrated into the cis side. Plotted are cis/trans $Ca^{2+}$ activity gradients (*Robinson and Stokes, 2002*) versus $E_{rev}$. Also plotted is the Nernst equation for calcium, which represents ideal selectivity for $Ca^{2+}$ over $Cl^-$ (**d**) $Ca^{2+}$ permeability in the presence of excess KCl. Before recording, the cis side was adjusted to pH 6.9 and then 1 µg APOL1 G0 was added to the cis side. APOL1 was allowed to associate with the bilayer for 1 hr and then the cis side was perfused with chamber buffer (150 mM KCl, 1 mM CaCl2, pH 7.2). Once recording began, the cis side was adjusted to pH 6.6, allowing for APOL1 insertion and channel formation. A large increase in the conductance upon re-neutralization of the cis side (pH 7.2) indicates pH-dependent channel opening. $E_{rev}$ (+1.75 mV) was determined by adjusting the voltage until the current read zero. $CaCl_2$ was then titrated into the cis compartment to the indicated concentrations. Upon each addition there was an upward shift in the current and the $E_{rev}$ became more negative, indicating $Ca^{2+}$ permeability of the APOL1 channel. The pCa/pK permeability ratio at 2 mM calcium was calculated as 0.6 (See Materials and methods).

*Thomson and Finkelstein, 2015*). To examine the ion selectivity of this conductance, we ascertained the reversal potential ($E_{rev}$, the voltage required to zero the current) before and after establishment of a 1.95-fold cis:trans $CaCl_2$ gradient. $E_{rev}$ became more negative after $CaCl_2$ addition, lowering from −1 mV to −6 mV, indicating selectivity for $Ca^{2+}$ over $Cl^-$ and demonstrating that rAPOL1 conducts $Ca^{2+}$ (*Figure 2b*). All rAPOL1 variants tested are equally permeable to $Ca^{2+}$ (*Figure 2c*), as well as $Na^+$ and $K^+$(*Thomson and Finkelstein, 2015*), indicating that a difference in ion selectivity is not the cause of disease.

We then tested whether APOL1 was measurably permeable to $Ca^{2+}$ at physiological salt concentrations (150 mM $K^+$, 1 mM $Ca^{2+}$). Adding an extra 1 mM $CaCl_2$ to the cis compartment caused a positive shift in the current and negative shift in the reversal potential (*Figure 2d*). The pCa/pK permeability ratio at 2 mM $Ca^{2+}$ was calculated as 0.6. This result confirms that APOL1 channels retain $Ca^{2+}$ permeability even in the presence of physiologically relevant KCl concentrations and suggests that APOL1 may lead to a cellular influx of $Ca^{2+}$.

To ascertain whether the RRVs lead to a cytoplasmic $Ca^{2+}$ influx upon induction, we transfected the cytoplasmic calcium indicator GCaMP6f (*Chen et al., 2013*) into FT293 cells. Performing live-cell microscopy with GCaMP6f and cell death marker DRAQ7 allowed us to determine the timing of events between a potential $Ca^{2+}$ influx and changes in cell morphology, plasma membrane integrity, and lysis. Upon induction, cells expressing the RRVs, but not G0, exhibited an increase in $Ca^{2+}$ beginning approximately 12–18 hr after induction (*Figure 3a*, *Video 1*). The $Ca^{2+}$ levels increased gradually and occurred several hours prior to cell swelling and membrane blebbing. Cells typically remained swollen for 12–18 hr before lysis (uptake of DRAQ7 was only detected in a few cells within 30 hr of induction, and only after lysis).

High-throughput microscopy was performed to analyze individual cells over time and compare the cell populations for changes in $Ca^{2+}$-dependent GCaMP6f fluorescence. There was no difference in mean GCaMP6f fluorescence between the variants without induction. However, after 30 hr of induction we observed a 2 to 3-fold increase in mean GCaMP6f fluorescence with G1 and G2 relative to G0 (*Figure 3b* and *Figure 3—Source data 1*). These data confirm that RRV expression leads to an increase of cytoplasmic $Ca^{2+}$ that precedes cell swelling and death, suggesting that $Ca^{2+}$ influx is an early event contributing to cytotoxicity.

## $Ca^{2+}$ influx and cytotoxicity require trafficking of RRVs out of the ER, and the source of $Ca^{2+}$ is extracellular

APOL1 contains a signal peptide (*Monajemi et al., 2002*) and localizes to the ER (*Cheng et al., 2015*) and PM (*O'Toole et al., 2018*; *Olabisi et al., 2016*), indicating passage through the secretory pathway. To investigate whether RRV-mediated cytotoxicity occurs when localized to the ER or by subsequent trafficking from it, we expressed the *APOL1* variants using the bicistronic RUSH plasmid (*Boncompain et al., 2012*). RUSH encodes streptavidin targeted to the ER lumen and streptavidin-binding-peptide tagged *APOL1* variants. The streptavidin binds to and retains tagged APOL1 in the ER. Upon treatment with biotin, APOL1 is released from the ER in a synchronous manner (*Figure 4a*).

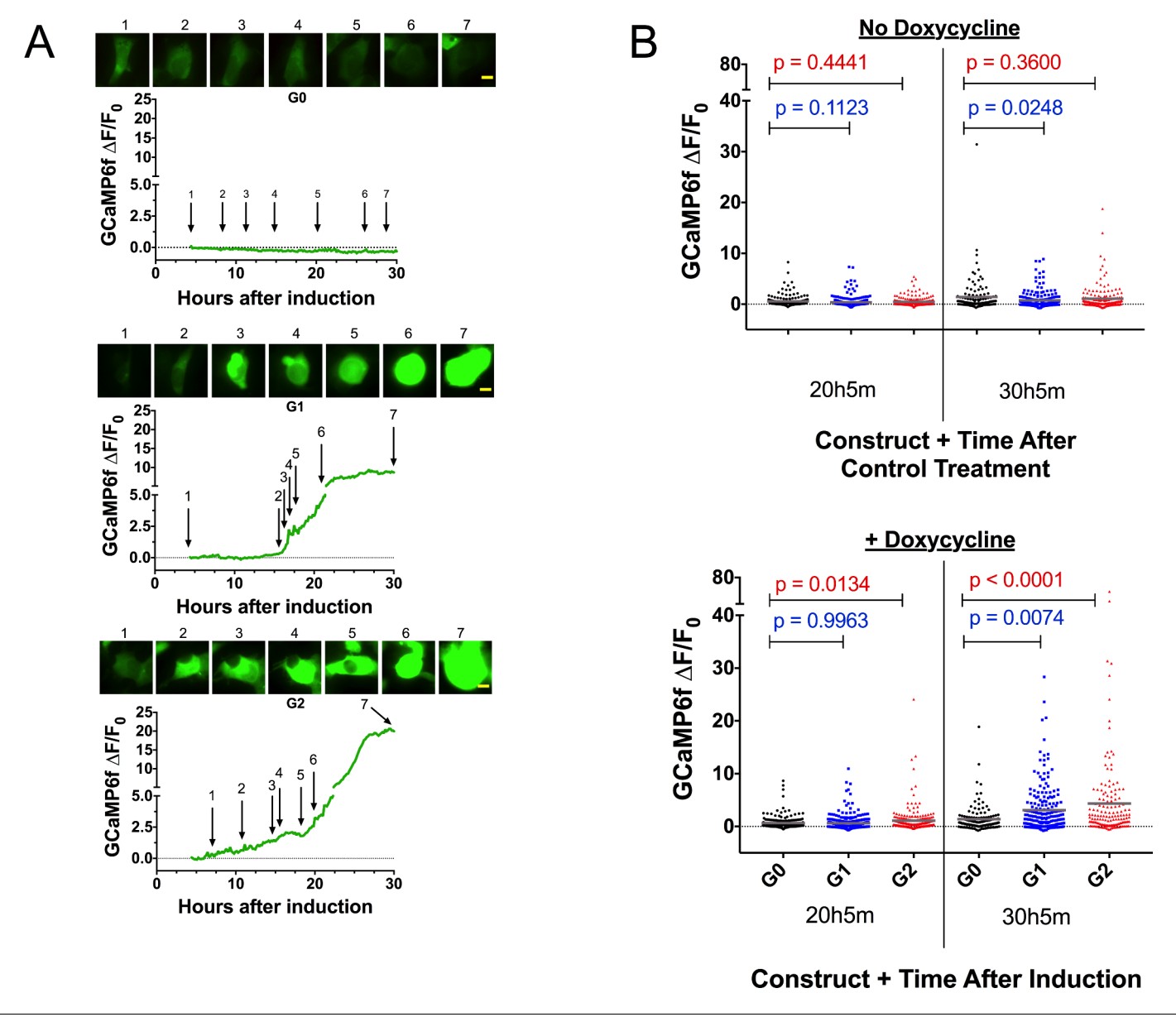

**Figure 3.** Expression of the RRVs leads to a $Ca^{2+}$ influx that precedes cell swelling and death. (a) Fluorescence traces of representative GCaMP6f-positive cells demonstrating that G1 and G2 cause a $Ca^{2+}$ influx prior to cell swelling. GCaMP6f-transfected FT293 cells were incubated with DRAQ7 followed by 50 ng/mL doxycycline to induce *APOL1* expression and then imaged via widefield every 10 min for 4.5–30 hr post induction. Traces represent levels of cytoplasmic $Ca^{2+}$ over time as measured by GCaMP6f fluorescence (no DRAQ7 was observed in depicted cells). Cells are from *Video 1*. Scale bars = 20 μm. (b) High-throughput analysis revealed a significant increase of cytoplasmic $Ca^{2+}$ levels driven by G1 and G2 compared to G0. Each point is the $\Delta F/F_0$ for an individually tracked cell and bars represent the cell population mean of GCaMP6f fluorescence. Cells were analyzed from 4 fields of view per condition, n = 1748. A one-way ANOVA multiple comparisons test was performed to compare the RRVs with G0 at the indicated timepoints.

The online version of this article includes the following source data for figure 3:

**Source data 1.** FT283 cells GCaMP6f microscopy, 30 hours after induction one way ANOVA.

RRV cytotoxicity required trafficking from the ER, leading to 20 % cell death 24 hr after biotin-mediated release in transfected HEK293 cells (*Figure 4b*). In contrast, G0 remained non-toxic after release. No cell death was detected when the APOL1 variants were retained in the ER. Cells producing RUSH-G0 and G1 variants exhibited similar levels of protein production after 24 hr of

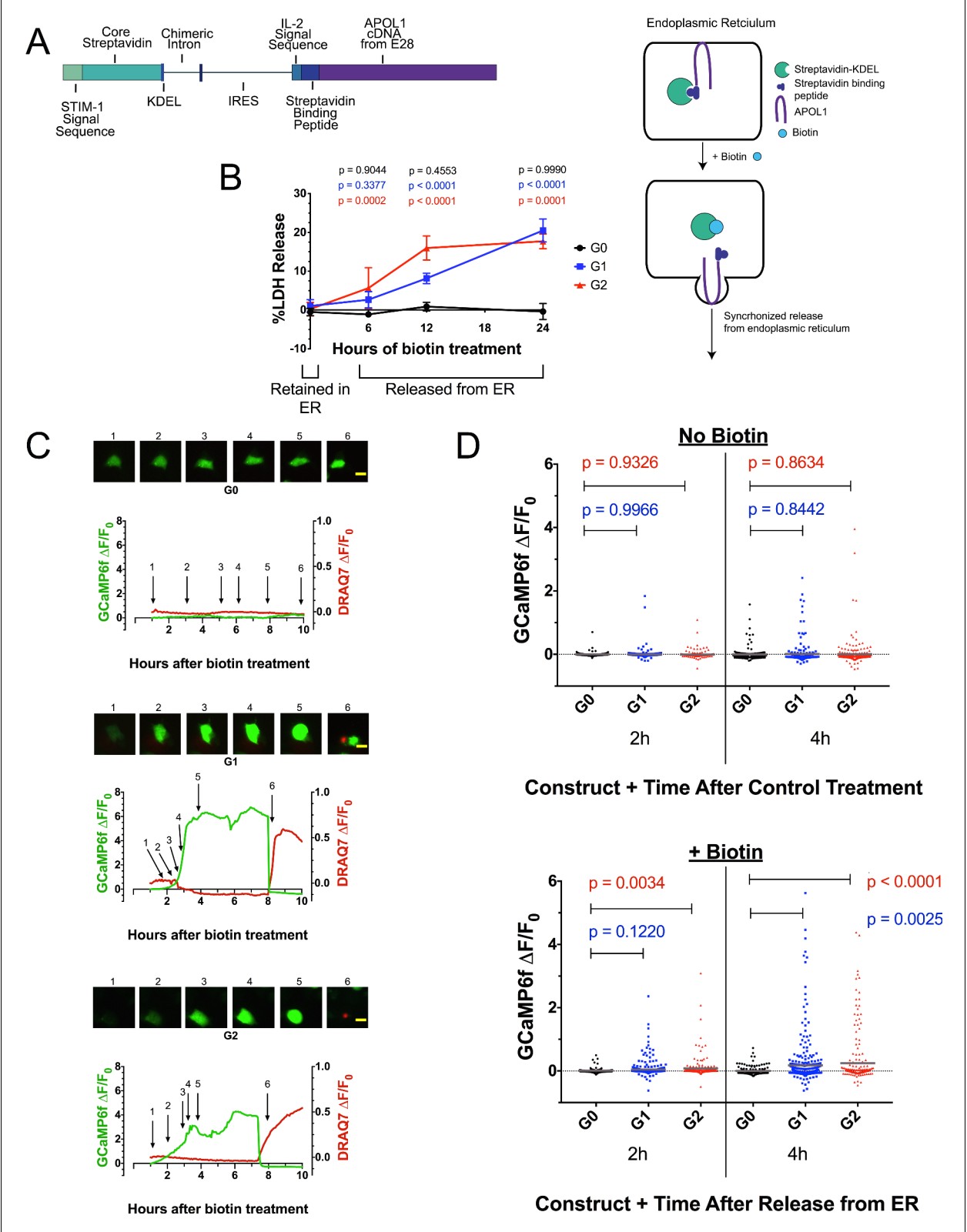

**Figure 4.** $Ca^{2+}$ influx and cytotoxicity of G1 and G2 requires trafficking from the ER. (a) Schematic of the RUSH system. Streptavidin was expressed with a signal peptide and KDEL allowing for localization and retention in the ER lumen along with streptavidin-binding protein (SBP) tagged APOL1. SBP binds to streptavidin causing APOL1 to be retained in the ER until synchronous release is initiated by the addition of biotin. (b) Time course showing that RRV cytotoxicity requires trafficking from the ER. 24 hr after transfection, HEK293 cells were treated with or without (0 hr) 80 µM biotin at the

*Figure 4 continued on next page*

*Figure 4 continued*

indicated times. 48 hr post-transfection, cytotoxicity was measured via release of lactate dehydrogenase. To compare cytotoxicity between biotin treated and untreated (0 hr) for respective genotypes, a two-way ANOVA with multiple comparisons was performed (n = 6). (c) Fluorescence traces of GCaMP6f-positive HEK293 cells showing that the G1 and G2-mediated $Ca^{2+}$ influx occurs after trafficking from the ER. GCaMP6f-transfected cells were incubated with DRAQ7 followed by 80 μM biotin to release APOL1 and were then imaged via widefield every 5 min for 1–18 hr post treatment. Cells are from *Video 2*. Scale bars = 20 μm. (d) High-throughput imaging and analysis was performed as in *Figure 3b*, demonstrating that the G1 and G2-mediated $Ca^{2+}$ influx requires trafficking from the ER. Each point is the $\Delta F/F_0$ for an individually tracked cell and bars represent the cell population mean of GCaMP6f fluorescence. Cells were analyzed from 3 fields of view per condition, n = 1657. A one-way ANOVA multiple comparisons test was performed to compare the RRVs with G0 at the indicated timepoints.

The online version of this article includes the following figure supplement(s) for figure 4:

**Figure supplement 1.** Validation of protein expression and $Ca^{2+}$-driven cytotoxicity of APOL1 in the RUSH system.
**Figure supplement 2.** The G1 and G2-mediated cytoplasmic $Ca^{2+}$ influx is not due to ER $Ca^{2+}$ release.

transfection, though RUSH-G2 expressed approximately 33% less protein (*Figure 4—figure supplement 1a*), possibly due to its higher cytotoxicity (*Figure 4b* at 12 hr, 4d at 2 hr with biotin).

We next utilized the RUSH system to determine if the previously observed RRV-mediated $Ca^{2+}$ influx also required exit from the ER by co-transfection with GCaMP6f. Biotin-mediated release of RUSH-G1 and G2 from the ER led to a rapid increase in cytoplasmic $Ca^{2+}$ within 2–4 hr of treatment. Approximately 2 hr after the initial $Ca^{2+}$ influx, membrane blebbing and cell swelling were observed. Cells remained swollen for 4–6 hr until lysis, after which DRAQ7 was detected (*Figure 4c*, *Video 2*).

High-throughput microscopy was performed revealing a significant increase in the cell population mean of GCaMP6f fluorescence between RUSH-G1 and G2 cells compared to G0. The analysis was limited to 4 hr post-biotin, as nearly all $Ca^{2+}$ influx had begun within that time frame. Without biotin, no difference in GCaMP6f fluorescence was detected. With biotin, RUSH-G1 cells displayed a significant increase in GCaMP6f fluorescence compared to RUSH-G0 at 4 hr, while an increase in RUSH-G2 could be detected as early as 2 hr post-release (*Figure 4d*, and *Figure 4—figure supplement 1b and e*). This experiment was reproduced in CHO cells (*Figure 4—figure supplement 1c–d*, *Video 3*). These results robustly demonstrate the requirement for G1 and G2 to exit the ER in order to drive a $Ca^{2+}$ influx and cytotoxicity.

The ER is the largest reservoir of intracellular $Ca^{2+}$ (*Burdakov et al., 2005*), and sequestration and release of ER $Ca^{2+}$ stores plays a pivotal role in many signaling and cell death pathways (*Berridge, 2002*; *Zhivotovsky and Orrenius, 2011*). To determine if ER $Ca^{2+}$ release occurs with APOL1 cytotoxicity, cells were co-transfected with RUSH-APOL1, GCaMP6f, and the ER $Ca^{2+}$ sensor ER-LAR-GECO (*Wu et al., 2014*). The combination of these sensors allows for visualization of ER $Ca^{2+}$ release or lack of re-uptake, as evidenced by treatment with the sarcoendoplasmic reticulum calcium transport ATPase inhibitor thapsigargin (*Figure 4—figure supplement 2a*). Cells exhibiting

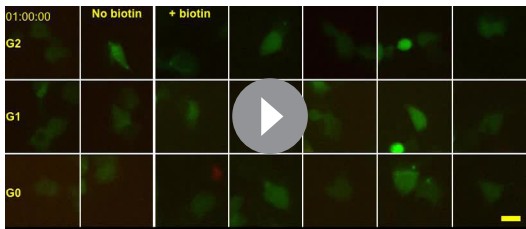

**Video 2.** Expression of RUSH-G1 and G2 leads to $Ca^{2+}$ influx, swelling, and lysis only after release from the ER. HEK293 cells were co-transfected with RUSH-APOL1 and GCaMP6f for 24 hr. Prior to imaging, 3 μM DRAQ7 was added and cells were treated with or without 80 μM biotin to release APOL1 from the ER. Cells were imaged via widefield from 1 to 18 hr post-biotin treatment, and dual color images were taken every 5 min. Scale bars = 20 μm.
https://elifesciences.org/articles/51185#video2

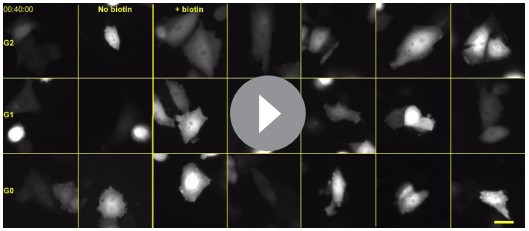

**Video 3.** Expression and ER release of RUSH-G1 and G2 leads to $Ca^{2+}$ influx and lysis in CHO cells. CHO cells were co-transfected with RUSH-APOL1 and GCaMP6f for 24 hr. Prior to imaging, cells were treated with or without 80 μM biotin to release APOL1 from the ER. Cells were imaged via widefield from 1 to 12 hr post biotin treatment, and images were taken every 5 min. Scale bars = 20 μm.
https://elifesciences.org/articles/51185#video3

the established phenotype of cytoplasmic $Ca^{2+}$ influx followed by swelling were analyzed for fluorescence changes in both sensors. While cytoplasmic $Ca^{2+}$ increases, there is no release of $Ca^{2+}$ from the ER (*Figure 4—figure supplement 2b–e*, *Video 4*). The lack of ER $Ca^{2+}$ release indicates that the source of $Ca^{2+}$ in RRV-mediated cytotoxicity is extracellular, possibly conducted via G1 and G2 cation channels at the PM.

## APOL1 localizes to the PM prior to $Ca^{2+}$ influx

Overexpressed APOL1 has previously been reported to reach the PM (*O'Toole et al., 2018*; *Olabisi et al., 2016*; *Heneghan et al., 2015*). We hypothesize that G1 and G2 must first localize to the PM in order to form cation channels that lead to the observed ion flux and cell swelling. Using confocal immunofluorescence microscopy, we tested whether RUSH-APOL1 would traffic to the PM after biotin treatment, and if localization to the PM occurs within a timeframe before $Ca^{2+}$ influx is first detected in CHO cells (1.75–2 hr post-release, *Figure 4—figure supplement 1d*, *Video 3*).

We found that RUSH-APOL1 traffics to the PM prior to detection of the $Ca^{2+}$ influx, consistent with APOL1 forming cation channels in the plasma membrane. Intracellular antibody staining of ER-retained RUSH-APOL1 cells reveals extensive co-localization with the calnexin-stained ER, as expected (*Figure 5a*, no biotin). After 90 min of biotin treatment, all three APOL1 variants are detected at the PM (*Figure 5a*, with biotin, white arrows). Additionally, APOL1 localizes to the peri-nuclear region post ER release, which is suggestive of localization within the Golgi or recycling endosomes after it exits the ER and traffics to the PM (*Figure 5—figure supplement 1*). Some RUSH-G1 and G2 expressing cells also undergo swelling after 90 min of biotin treatment. In swollen cells, the PM is enriched in APOL1 and the ER is retracted, potentially due to hydrostatic pressure.

Correspondingly, via cell surface immunostaining, all RUSH-APOL1 variants were detected at the PM within 90 min of release and displayed a punctate staining pattern (*Figure 5b*). Due to the leakiness of the RUSH system, APOL1 was also found at the PM of some untreated cells. However, high-throughput microscopy of the transfected cells treated with biotin for 0–120 min revealed a steady increase of APOL1 localization to the PM, peaking at 90 min post-release. After 90 min of biotin treatment, mean APOL1 signal intensity at the PM increased 25–30%, and the number of cells positive for APOL1 staining at the cell surface increased 3–4 fold compared to untreated cells (*Figure 5c–e*). Less G2 was detected at the surface compared to G0 and G1 (*Figure 5b–c*), potentially due to the combination of lower protein expression (*Figure 4—figure supplement 1a*) and higher cytotoxicity (*Figure 4b* at 12 hr and 4d at 2 hr with

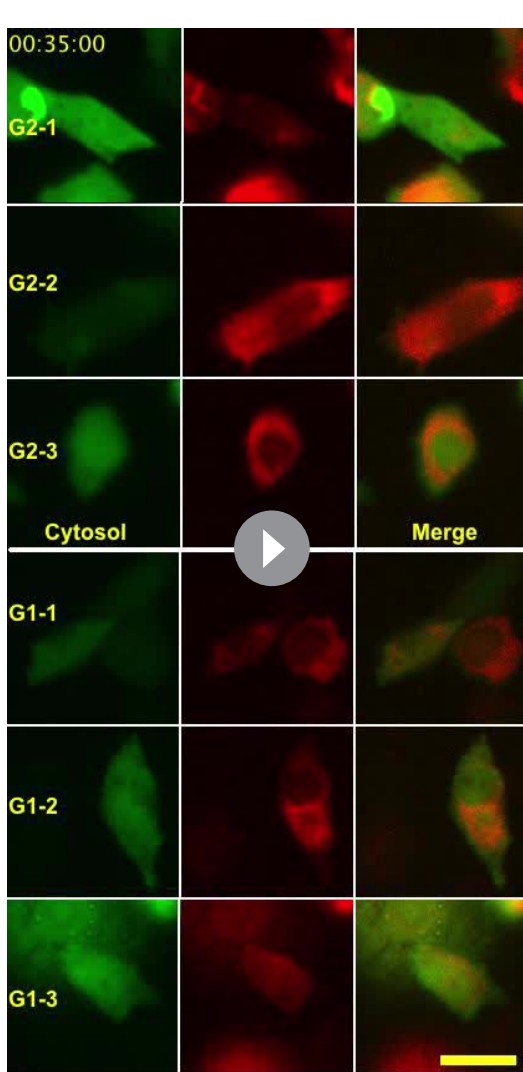

**Video 4.** Expression and release of RUSH-G1 and G2 does not induce ER $Ca^{2+}$ release. CHO cells were co-transfected with RUSH-APOL1, GCaMP6f, and ER-LAR-GECO for 24 hr prior to imaging. On the day of the experiment cells were treated with 80 μM biotin and imaged for 0.5–12 hr post treatment. Cells that displayed the established phenotype of $Ca^{2+}$ influx followed by cell swelling were selected. Dual color images were taken every 5 min. Scale bars = 20 μm.
https://elifesciences.org/articles/51185#video4

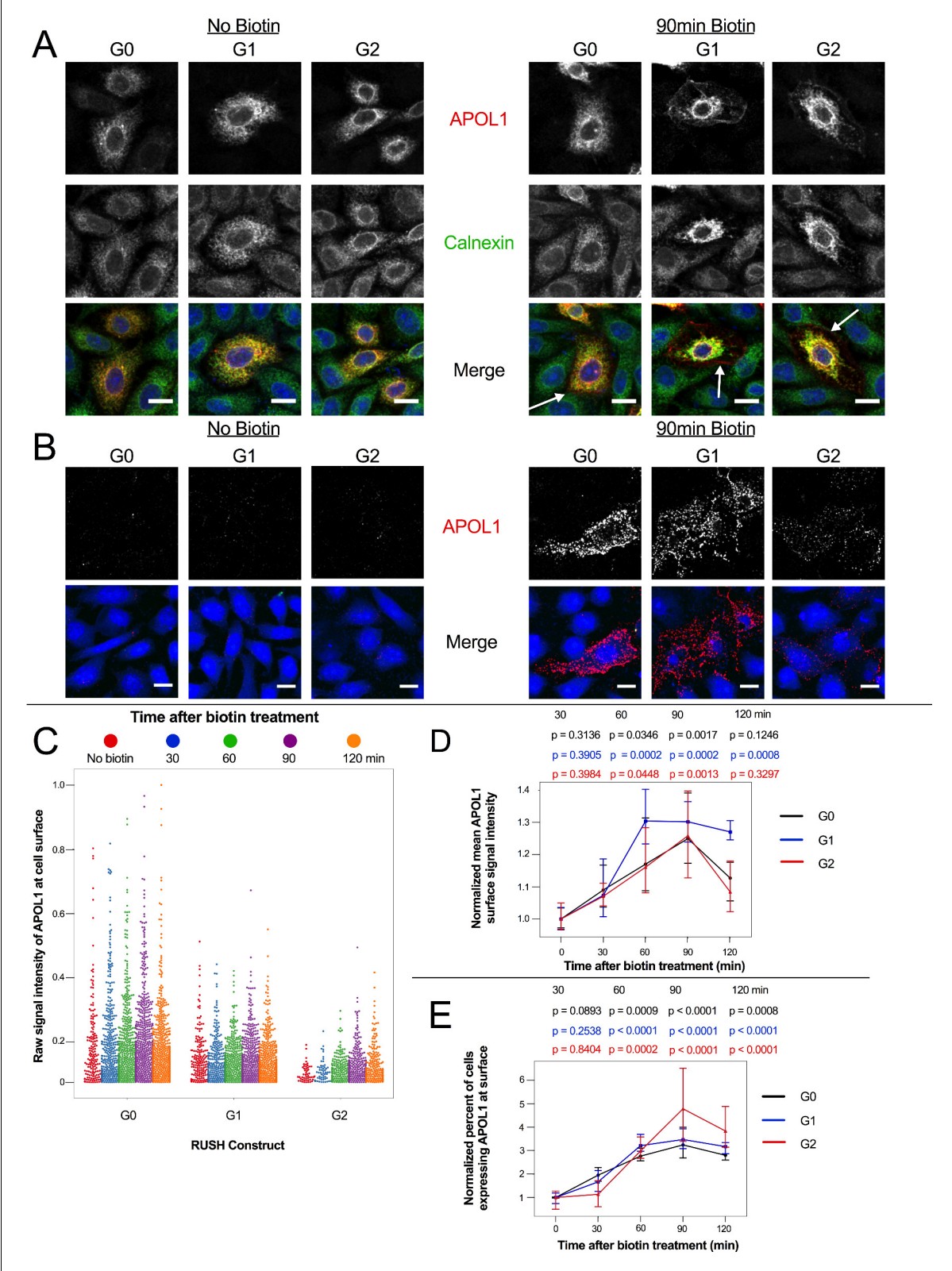

**Figure 5.** APOL1 traffics to the PM prior to Ca²⁺ influx. (**a**) Confocal images of transfected and permeabilized CHO cells depict RUSH-APOL1 (red) localized to the ER (stained via calnexin, green) without biotin followed by partial PM localization after 90 min of biotin treatment. Representative cells from n = 3 independent experiments. (**b**) RUSH-APOL1 localizes to and forms punctae at the PM within 90 min of biotin treatment. CHO cells were treated and imaged as in (**a**) except without permeabilization. Here anti-calnexin (green) was used as a control for cell permeabilization (depicted in the

*Figure 5 continued on next page*

Figure 5 continued

merged images, no permeabilization was detected). Representative cells from n = 4 independent experiments. (c) High-throughput confocal microscopy reveals that RUSH-APOL1 begins localizing to the PM within 60–90 min. Cells were randomly imaged at 20x, capturing ≥10 fields of view per well from 3 replicate wells for each condition. Calnexin signal was used to filter out permeabilized cells. Each dot represents a single cell (n = 462,918 cells analyzed). (d–e) RUSH-APOL1 localization to the PM steadily increases until 90 min post release from the ER. (d) The mean intensity of all cells in (c) was normalized to the respective no biotin controls. (e) The percentage of cells expressing RUSH-APOL1 at the PM was determined using a threshold set by untransfected wells, and then normalized to the respective no biotin controls. For analysis of (d) and (e), a generalized linear model was used to make pairwise comparisons between all samples. Comparisons were performed between biotin treated and untreated cells within each respective genotype. All data are represented as mean ± s.d. (a–b) Scale bars = 10 μm.

The online version of this article includes the following figure supplement(s) for figure 5:

**Figure supplement 1.** RUSH-APOL1 traffics to the peri-nuclear region and PM post-biotin treatment.

biotin). However, cell surface expression still increased in a similar manner compared to G0 and G1. These results demonstrate that RUSH-APOL1 traffics to the PM within the timeframe that a cytoplasmic increase in $Ca^{2+}$ is first detected, and suggests that G1 and G2 form cation channels at the PM as an early event that leads to cytotoxicity.

## RRV-mediated cytotoxicity is driven by the influx of both $Na^+$ and $Ca^{2+}$

As a non-selective cation channel, APOL1 may lead to cell death in a variety of ways. It has been postulated that the driver of cell death is APOL1-mediated $K^+$ efflux (*Olabisi et al., 2016*). In that study, Olabisi et al. incubated *APOL1*-expressing 293 cells in 'CKCM' media for 24 hr, in which all $Na^+$ was replaced by $K^+$. The study reported that incubating cells in CKCM reduced RRV cytotoxicity by approximately 50%. Additionally, APOL1 led to $K^+$ efflux in trypanosomes (along with a $Ca^{2+}$ influx) (*Rifkin, 1984*). While APOL1 undoubtedly leads to a $K^+$ efflux, the cell is already highly permeable to $K^+$ due to the presence of leak channels in the plasma membrane, allowing the cell to rapidly respond to changes in membrane potential or cell volume. Conversely, the cell membrane is minimally permeable to $Na^+$ and $Ca^{2+}$, and this permeability could significantly increase in the presence of open G1 and G2 channels at the cell surface. Therefore, we hypothesized cytotoxicity is driven by the influx of $Na^+$ and $Ca^{2+}$, rather than solely by the efflux of $K^+$.

We sought to replicate the conditions of the CKCM experiment performed by Olabisi et al. In addition to replacement of $Na^+$ with $K^+$, we also tested $Na^+$ replacement with the larger choline$^+$. rAPOL1 channels were tested for permeability of choline$^+$ in the planar lipid bilayer system (*Figure 6a*). Under conditions of symmetrical 150 mM KCl, $E_{rev}$ was +1 mV (*Figure 6b*), and when the cis-side was perfused and replaced with buffer containing 150 mM NaCl (leaving trans 150 mM KCl unchanged), there was only a slight change in $E_{rev}$ to −2 mV. However, when cis NaCl was perfused and replaced with choline Cl (trans KCl unchanged), there was a significant increase in $E_{rev}$ to +60 mV, indicating conductance of $K^+$ from the trans to the cis side, and minimal conductance of choline$^+$ (*Figure 6b*). Substituting into the Goldman-Hodgkin-Katz equation (assuming zero permeability to chloride) gives K:Na and K:choline permeability ratios of 1.0:1.1 and 1.0:0.1 respectively. These results demonstrate that the APOL1 channel is at least 10 times more permeable to $K^+$ and $Na^+$ than choline$^+$.

In order to replace $Na^+$ with $K^+$ or choline$^+$ in the cell culture media, we first tested the effect these conditions have on cell viability. HEK cells were transfected with RUSH-G0 plasmid and incubated for 12 hr in HBSS containing various amounts of NaCl replaced with equal amounts of choline Cl or KCl to make up 130 mM of salt (media contained 150 mM $Na^+$). Reducing $Na^+$ to 52.5 mM and replacing it with $K^+$ led to a 40% reduction in cell viability, whereas replacement with choline$^+$ only had a modest effect on cell viability (*Figure 6—figure supplement 1a*). Further reduction of $Na^+$ to 20 mM led to a 75% decrease in viability when replaced by choline+ (*Figure 6—figure supplement 1b*). Due to these results, we decided to lower $Na^+$ to 85 mM for further experimentation, where choline$^+$ had no significant effect on viability. A loss of cell viability with $K^+$ replacement of $Na^+$, however, was unavoidable even at higher concentrations of $Na^+$. Additionally, higher amounts of KCl were avoided due to its ability to depolarize the cell.

Reduction of $Na^+$ to 85 mM inhibited RRV cytotoxicity by 40–60% (*Figure 6c*). This level of rescue was similar to the amount reported by Olabisi et al. There was no significant difference between

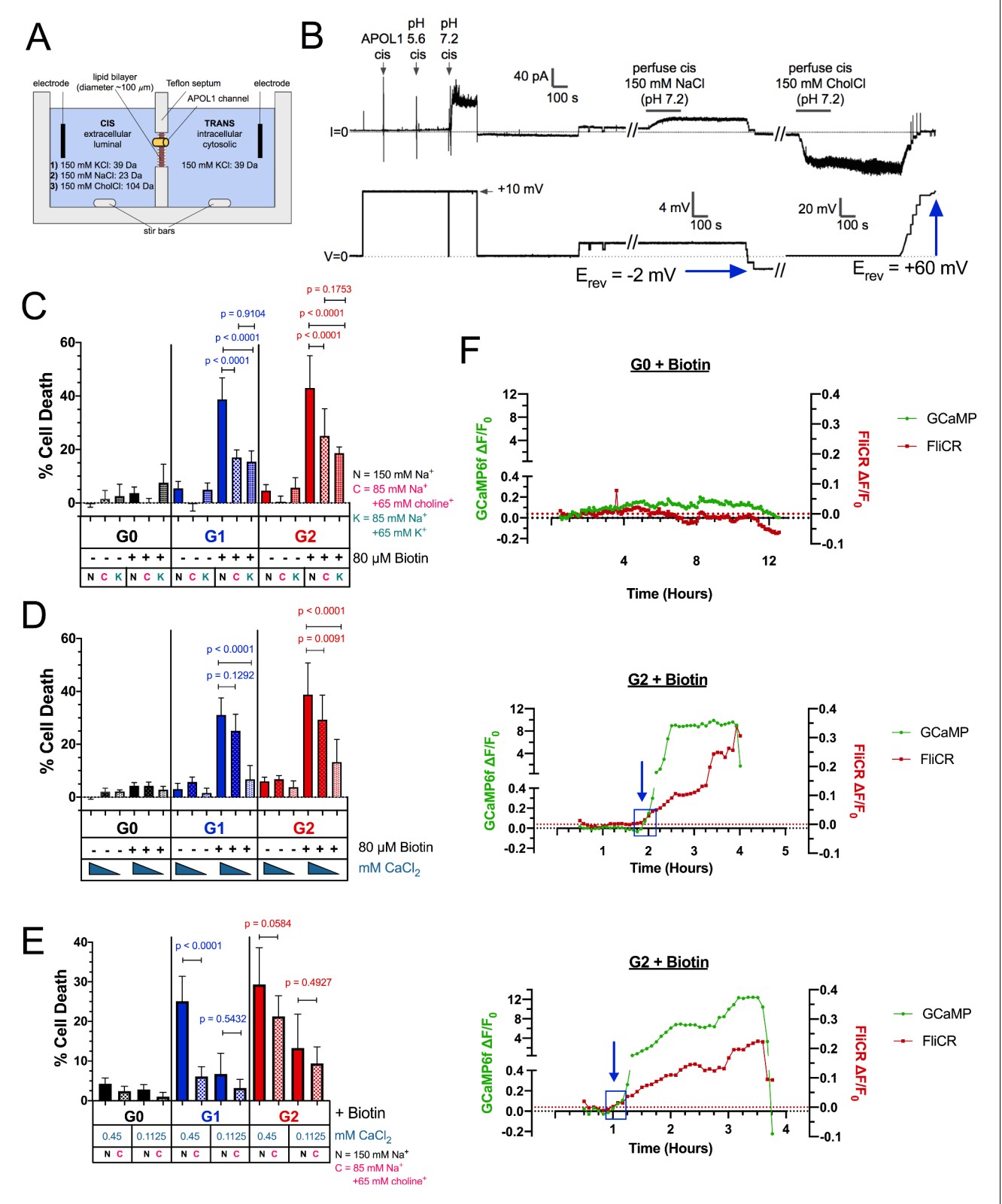

**Figure 6.** RRV cytotoxicity is driven by the influx of both $Na^+$ and $Ca^{2+}$. (a) Schematic of the planar lipid bilayer setup showing the sequence of cis buffer perfusions. (b) The APOL1 channel is readily permeable to $Na^+$, but not choline$^+$. In symmetrical KCl solutions $E_{rev}$ was determined as + 1 mV. Then, after cis perfusion with equimolar NaCl buffer (pH 7.2, horizontal bar) there was only a slight change in $E_{rev}$ ($E_{rev}$ = −2 mV; 4 mV scale). In contrast, $E_{rev}$ increased to + 60 mV after exchanging the cis solution for chamber buffer containing equimolar choline chloride. Substituting into the

*Figure 6 continued on next page*

Figure 6 continued

Goldman-Hodgkin-Katz equation (assuming zero permeability to chloride) gives K:Na and K:choline permeability ratios of 1.0:1.1 and 1.0:0.1 respectively. There are two breaks in the record (indicated by //), during which the perfuser was recharged with the appropriate solution. (c) The cytotoxicity of the RRVs in RUSH transfected HEK293 cells is significantly reduced by lowering extracellular $Na^+$ from 150 mM to 85 mM. The rescue from cytotoxicity was indistinguishable between replacement with either $K^+$ or choline$^+$ (n = 9). (d) RRV cytotoxicity was reduced by lowering extracellular $Ca^{2+}$ from 1.8 mM to 0.45 or 0.1125 mM (n = 12). (e) Reduction of both extracellular $Ca^{2+}$ and $Na^+$ (replaced by choline$^+$) has an additive effect in lowering RRV cytotoxicity, as seen by further rescue from cell death with 0.45 mM $Ca^{2+}$ combined with 85 mM $Na^+$ (n = 13). (c–e) Cell death was assayed 12 hr post-biotin treatment with the Promega MultiTox fluorescent assay. Two-way ANOVAs with multiple comparisons were performed. (f) RRV mediated cytotoxicity is driven by the concurrent influx of both $Ca^{2+}$ and $Na^+$. CHO cells were co-transfected with either RUSH-G0 or G2, GCaMP6f, and the membrane voltage sensor FliCR. G2 cells exhibiting the established phenotype of $Ca^{2+}$ influx followed by cell swelling were analyzed for changes in membrane voltage (used as a surrogate for the influx of $Na^+$) (n = 21). G0 cells treated with biotin were analyzed for comparison (n = 7). Blue boxes and arrows indicate when the sustained increase in $Ca^{2+}$ initiates. The representative cells from this figure can be viewed in *Figure 6—video 1*.

The online version of this article includes the following video and figure supplement(s) for figure 6:

**Figure supplement 1.** Replacement of $Na^+$ with $K^+$ significantly reduces cell viability.
**Figure supplement 2.** Validation of the FliCR sensor.
**Figure supplement 3.** The RUSH-G2-mediated $Ca^{2+}$ influx occurs concurrently with a modest depolarization of the cell ($Na^+$ influx), followed by complete depolarization prior to cell death.
**Figure 6—video 1.** The G2-mediated $Ca^{2+}$ influx occurs concurrently with the influx of $Na^+$.
https://elifesciences.org/articles/51185#fig6video1

rescue caused by replacement with choline$^+$ or $K^+$, suggesting that the influx of $Na^+$ is a driver of APOL1-mediated cell death, and that it is upstream of the previously reported $K^+$ efflux.

As G1 and G2 lead to a cellular influx of $Ca^{2+}$, we tested whether $Ca^{2+}$ itself may act as a driver of cell death. Extracellular $Ca^{2+}$ was serially diluted from 1.8 mM to 0.1125 mM, and a significant reduction in cell death was recorded at $Ca^{2+}$-concentrations $\leq$ 0.45 mM, (*Figure 6d* and *Figure 6—figure supplement 1c*). Conversely, increasing extracellular $Ca^{2+}$ exacerbated cell death (*Figure 6—figure supplement 1d–e*). Reduction of $Ca^{2+}$ and $Na^+$ simultaneously (0.45 mM $Ca^{2+}$ and 85 mM $Na^+$ supplemented with choline$^+$) had an additive effect on inhibiting RRV cytotoxicity (*Figure 6e*). These results demonstrate that both $Na^+$ and $Ca^{2+}$ influx are the initial drivers of RRV-mediated cell death.

Having demonstrated a role for extracellular $Na^+$ and $Ca^{2+}$ in RRV-induced cell death, we next sought to measure $Na^+$ influx and compare its timing with the influx of $Ca^{2+}$. Due to a lack of any genetically encoded $Na^+$ sensors for long-term imaging, we utilized the plasma membrane voltage sensor FliCR as a readout for $Na^+$ influx. FliCR expressing cells exhibited an approximately 20–30% increase in $\Delta F/F_0$ upon depolarization with addition of 50 mM KCl to the extracellular milieu (*Figure 6—figure supplement 2a*).

CHO cells were co-transfected with RUSH G0 or G2, GCaMP6f, and FliCR and imaged every 5 min for 0.5–12 hr after biotin treatment. No sustained increases in cytoplasmic $Ca^{2+}$ or membrane depolarization were detected in G0-expressing cells (*Figure 6f* and *Figure 6—figure supplement 2b*). The initial G2-mediated $Ca^{2+}$ influx occurred concurrently with an increase in

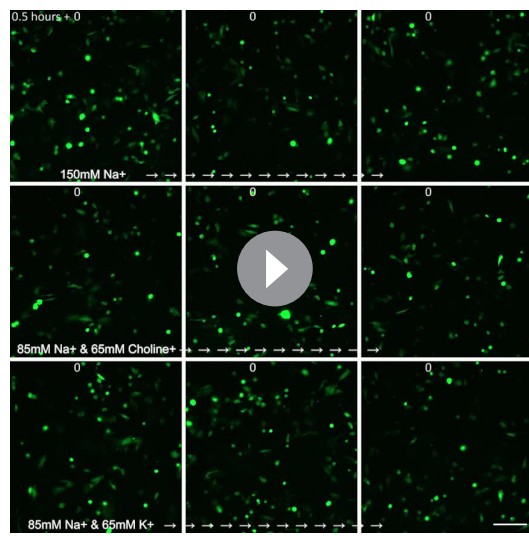

**Video 5.** Replacement of NaCl with choline Cl or KCl does not affect the G2-mediated $Ca^{2+}$ influx. CHO cells were co-transfected with RUSH-G2 and GCaMP6f for 24 hr prior to imaging. On the day of the experiment cells were treated with 80 µM biotin and incubated in media containing 150 mM $Na^+$ (130 mM NaCl), 85 mM $Na^+$ and 65 mM choline$^+$, or 85 mM $Na^+$ and 65 mM $K^+$. The G2-mediated $Ca^{2+}$ influx was unaffected by reduced $Na^+$. 3 different fields are shown for each condition. Cells were imaged every 5 min from 0.5 to 12 hr post-biotin. Scale bar = 100 µm.
https://elifesciences.org/articles/51185#video5

membrane depolarization (*Figure 6f* blue boxes, *Figure 6—figure supplement 3*, *Figure 6—video 1*). As $Ca^{2+}$ continued to accumulate within the cytoplasm, cells either exhibited a parallel increase in membrane depolarization, or underwent erratic fluctuations followed by significant depolarization prior to cell death (*Figure 6—figure supplement 3*).

$Na^+$ influx can drive accumulation of intracellular $Ca^{2+}$, either by disrupting the $Na^+/Ca^{2+}$ exchanger at the PM (*Barry et al., 1985*) or by depolarizing the cell and causing voltage-gated $Ca^{2+}$ channels to open. However, while these events may contribute to $Ca^{2+}$ accumulation, APOL1 itself can conduct $Ca^{2+}$ (*Figure 2a–d*) and leads to a large and sustained influx. Indeed, the G2-mediated $Ca^{2+}$ influx was unaffected by $Na^+$ replacement with choline$^+$ or $K^+$ (*Video 5*). Therefore, the RRVs drive cytotoxicity by directly increasing the membrane permeability of both $Na^+$ and $Ca^{2+}$.

## The cytotoxicity of all APOL1 variants is dependent upon acid-driven activation

The APOL1 cation channel requires two steps to become functional: an acidic pH to drive irreversible membrane insertion, followed by a neutral pH to open the channel (*Figure 2b*; *Thomson and Finkelstein, 2015*). Acidification is also required for trypanolytic activity, as trypanosomes pretreated with the weak base ammonium chloride are protected against APOL1 (*Hager et al., 1994*). Within a mammalian cell, APOL1 can encounter acidic and neutral environments by trafficking along the secretory pathway (*Paroutis et al., 2004*). However, while all three APOL1 variants traffic to the PM and form channels that are permissive to $Na^+$, $K^+$ (*Thomson and Finkelstein, 2015*), and $Ca^{2+}$ (*Figure 2c*) in a planar lipid bilayer, only G1 and G2 lead to cytotoxicity in our models and cause disease. The existence of a chaperone in mammalian cells that mimics the serum resistance associated protein (SRA) (*Limou et al., 2015*) found in *Trypanosoma brucei rhodesiense* (*Pérez-Morga et al., 2005*) has been proposed. SRA directly binds to and inactivates G0, preventing its acid activation, but is evaded by G2 (*Thomson and Finkelstein, 2015*; *Thomson et al., 2014*). If G0 is sequestered by an unknown chaperone while trafficking along the secretory pathway, or alternatively if G0 is less sensitive to pH changes relative to the RRVs, this could prevent the insertion event necessary for channel formation.

To circumvent this potential regulatory mechanism along the secretory pathway, RUSH-APOL1 transfected cells were transiently acidified at pH 5.5 and then re-neutralized. This was performed 2 hr post biotin-mediated release, allowing for APOL1 localization to the PM. Under these conditions, RUSH-G0 led to 12.5 % cytotoxicity after release from the ER (*Figure 7a*). Additionally, the cytotoxicity of RUSH-G1 and G2 increased 1.5 to 2-fold if acidified. Conversely, reducing the acid-activation of APOL1 by pre-treatment with the weak base ammonium chloride significantly lowered the cytotoxicity of G1 and G2 (*Figure 7b*). The modulation of APOL1-mediated cell death by raising or lowering the pH indicates that not all G1 and G2 at the PM are in an active channel state. These results demonstrate that G0 contains the potential to be innately cytotoxic, however this cytotoxicity is prevented by an unknown mechanism. Conversely, G1 and G2 more readily convert into the active channel state during periods of sustained expression, leading to cell death and disease (*Figure 8*).

## Discussion

The goal of this study was to investigate the underlying mechanisms driving APOL1-mediated kidney disease. A comprehensive analysis using genetic, biochemical, and microscopy-based approaches revealed that RRV-mediated cytotoxicity first requires trafficking out of the ER to the PM, where they cause a cytotoxic cation flux followed by cell swelling, culminating in lysis. As channel activity leading to a $Ca^{2+}$ and $Na^+$ influx is the earliest observed event leading to cell death, and because $Ca^{2+}$ is a potent signaling molecule that can activate many signaling and cell death pathways (*Berridge, 2002*; *Zhivotovsky and Orrenius, 2011*), we propose this upstream event links the many APOL1-associated cell death pathways together.

We first replicated previously reported results that the RRVs lead to cytotoxicity (*Figure 1c*; *Olabisi et al., 2016*), which was marked by membrane blebbing and a swollen cell phenotype (*Videos 1–3*), the latter of which was also observed in human serum treated trypanosomes (*Rifkin, 1984*). Importantly, we used the naturally occurring alleles of *APOL1* (*Figure 1a*). A study by O'Toole et al. reported that G0 and the RRVs were equally cytotoxic, however their approached utilized artificially synthesized RRVs where C-terminal mutations were introduced into the G4 allele (*Figure 1a*;

*O'Toole et al., 2018*), which has been shown to have reduced cytotoxicity compared to the naturally occurring haplotype (*Lannon et al., 2019*). As the lytic activity of APOL1 is sensitive to even single amino acid changes (*Cuypers et al., 2016*), it is imperative to only use the naturally occurring alleles to draw relevant conclusions regarding kidney disease. While we have used the most prevalent haplotype of G0 with amino acid K150 as our control, it is important to consider the use of G0 E150 for future studies, as G1 and G2 arose in the E150 haplotype background (*Figure 1a*). It should be noted that there was no difference in cytotoxicity between G0 E150 and G0 K150 as reported by Lannon et al.

We are the first to report that the APOL1 channel is permeable to $Ca^{2+}$, however the selectivity of APOL1 has been controversial. $Cl^-$ selectivity was first reported, however this study utilized a truncated rAPOL1 (*Pérez-Morga et al., 2005*) that was later shown to be non-functional (*Molina-Portela et al., 2008*). $Cl^-$ selectivity was also reported using KCl-loaded large unilamellar vesicles (*Bruno et al., 2017*), however, $Cl^-$ selectivity only occurred at pH 5.0, and they reported APOL1 was $K^+$ selective at pH 7.1. While APOL1 may indeed be permeable to $Cl^-$, this occurs at pH 5.0 where only a minor current is recorded in planar lipid bilayers. Once neutralized, the current increases several hundred-fold due to the opening of the APOL1 cation channels (*Figure 2b*; *Thomson and Finkelstein, 2015*). Opening of these cation channels (but not the minor conductance at acidic pH) was also inhibited by recombinant SRA protein, suggesting a relevance to trypanosome lysis (*Thomson and Finkelstein, 2015*). Additionally, APOL1 causes the dissipation of $Na^+$ (*Figure 6f*) and $K^+$ gradients in animal cells and trypanosomes (*Rifkin, 1984*; *Olabisi et al., 2016*; *Heneghan et al., 2015*), and its trypanolytic activity is inhibited when extracellular $Na^+$ is replaced by larger cations that are APOL1-impermeant (*Molina-Portela et al., 2005*; *Figure 6a–b*). The results in this paper, together with the previous studies strongly implicate a role for the APOL1 cation channel in biological function, whereas any relevance of $Cl^-$ flux remains to be demonstrated.

Through the use of live-cell microscopy with $Ca^{2+}$ sensors GCaMP6f and ER-LAR-GECO and the membrane voltage sensor FliCR, we were able to discern that the upstream event leading to cell death is a cytoplasmic influx of extracellular $Na^+$ and $Ca^{2+}$. This was robustly reproduced in multiple cell lines and with the RUSH system, revealing that trafficking of APOL1 out of the ER was required for cytotoxicity. While we described an APOL1-driven $Ca^{2+}$ influx in this study, another group reported no changes in cytoplasmic $Ca^{2+}$(*O'Toole et al., 2018*). However, their approach utilized the dye Fura-2 measured via fluorescent plate reader at a single timepoint. Using the genetically encoded GCaMP6f and the more sensitive technique of time-lapse, live-cell fluorescent microscopy allowed us to observe the robust $Ca^{2+}$ influx caused by the RRVs.

$K^+$ efflux has been proposed as the mechanism that drives APOL1-mediated cell death, however our data demonstrate that this event is likely a response to $Na^+$ influx. Unlike $K^+$, $Na^+$ and $Ca^{2+}$ have a very low permeability across the plasma membrane meaning that APOL1 channels reaching the cell surface will conduct a measurable influx of $Na^+$ and $Ca^{2+}$. $Na^+$ influx will lead to the observed swelling and depolarization, both of which cause the classic cellular response of $K^+$ efflux, which in turn can also affect cell viability via stress activated protein kinases as previously reported (*Olabisi et al., 2016*). Additionally, the observed protection offered by CKCM media (replacement of all $Na^+$ with $K^+$) is likely due to the fact that $Na^+$ has been removed (*Figure 6c*). As APOL1 is also permeable to $K^+$, it likely contributes directly to the efflux of $K^+$, though not to the same degree as $Na^+$ and $Ca^{2+}$ influx due to of the presence of other $K^+$ channels at the PM. Independent of $Na^+/K^+$ flux, a sustained influx of $Ca^{2+}$ over several hours, which is usually tightly regulated by the cell, can activate a multitude of signaling pathways that may also contribute to cell death.

In order for G1 and G2 to drive a cytotoxic influx of extracellular $Ca^{2+}$, we reasoned that they must reach the PM within the 120 min where an increase of cytoplasmic $Ca^{2+}$ is first detected. Indeed, high-throughput confocal immunofluorescence revealed that all variants of RUSH-APOL1 trafficked with similar kinetics to the PM, which occurred within 60–90 min of biotin treatment (release of ER-retained APOL1). Upon reaching the PM, APOL1 formed a punctate staining pattern. Additionally, localization of APOL1 to the PM was corroborated by other studies (*O'Toole et al., 2018*; *Olabisi et al., 2016*; *Heneghan et al., 2015*). We also report that APOL1 localizes to the perinuclear region in some cells, indicating passage through the Golgi and/or endosomes. Interestingly, we observed that the ER had receded from its association with the PM in swollen cells, potentially due to an increase in hydrostatic pressure. A similar phenotype was observed in human submandibular gland cells treated with a hypotonic solution, which caused a loss of ER:PM contact sites

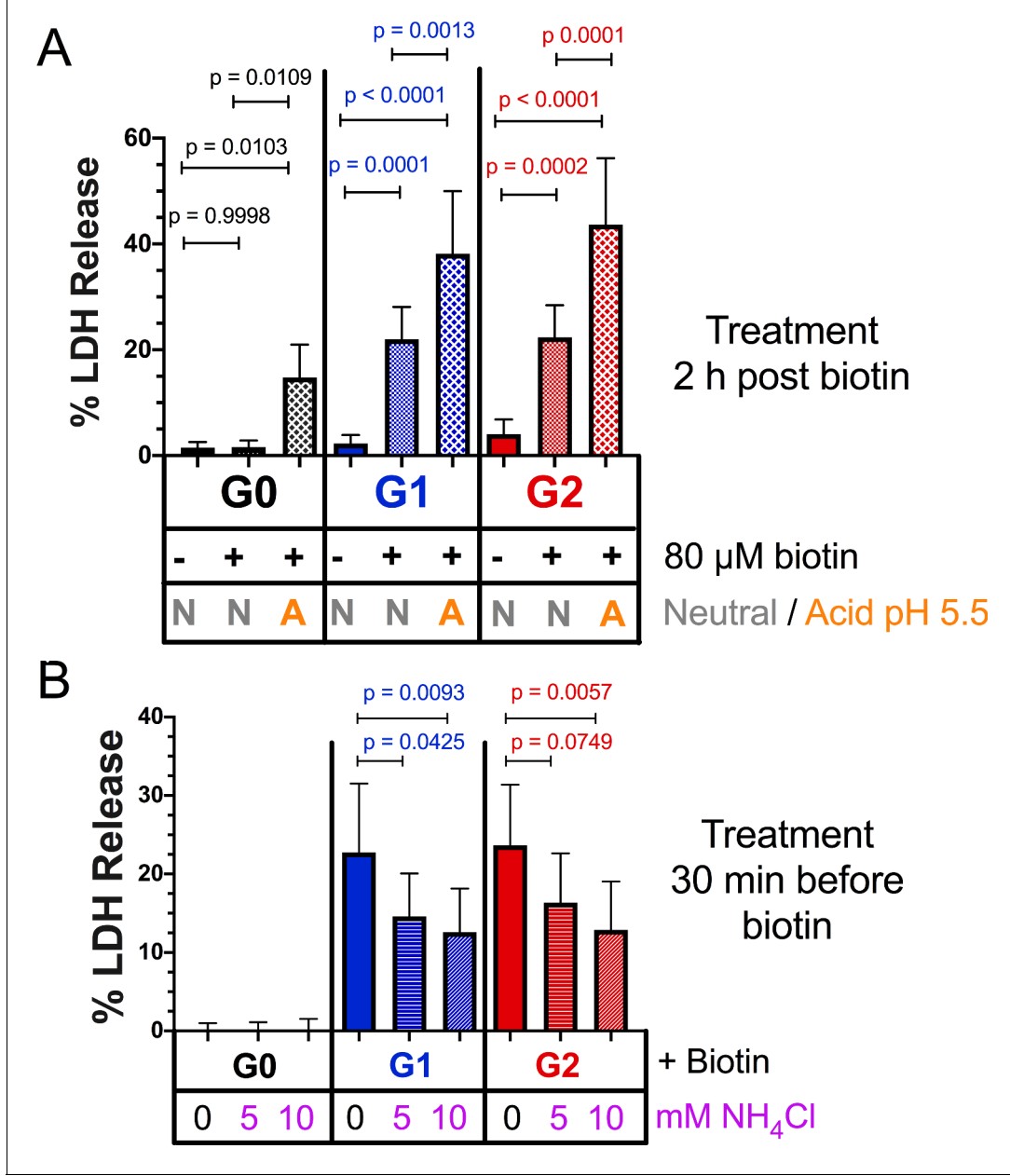

**Figure 7.** Acidic activation of APOL1 drives channel formation and cytotoxicity. (**a**) Acidification and neutralization of RUSH-APOL1 transfected HEK293 cells causes G0 to become cytotoxic and exacerbates the cytotoxicity of G1 and G2. 24 hr after transfection cells were treated with or without 80 μM biotin. 2 hr post-biotin, cells were incubated with media +/- succinic acid at pH 5.5 for 1 hr followed by neutralization. Cytotoxicity was measured 24 hr post-biotin (n = 13). (**b**) Pre-treatment with ammonium chloride protects against the cytotoxicity of G1 and G2. RUSH-APOL1 transfected HEK293 cells were treated with the indicated amounts of ammonium chloride 30 min prior to biotin treatment. Cytotoxicity was then measured 8 hr after biotin-mediated release (n = 11). (**a–b**) Cytotoxicity was measured via release of lactate dehydrogenase. A two–way ANOVA comparing treated and untreated cells within each respective genotype was performed.

(*Liu et al., 2010*). This may help explain the previously reported case of APOL1-mediated ER stress (*Wen et al., 2018*).

While all three variants are equally permissive for $Ca^{2+}$ and traffic to the PM in a similar fashion, only the RRVs are cytotoxic. It has been postulated that an SRA-like chaperone may exist to prevent G0 cytotoxicity (*Limou et al., 2015*), however no such binding partner has been found. We discovered that by artificially acidifying and therefore activating the non-toxic G0 after PM-localization, followed by re-neutralization, G0 led to cell death. Under these conditions RRV-mediated cytotoxicity

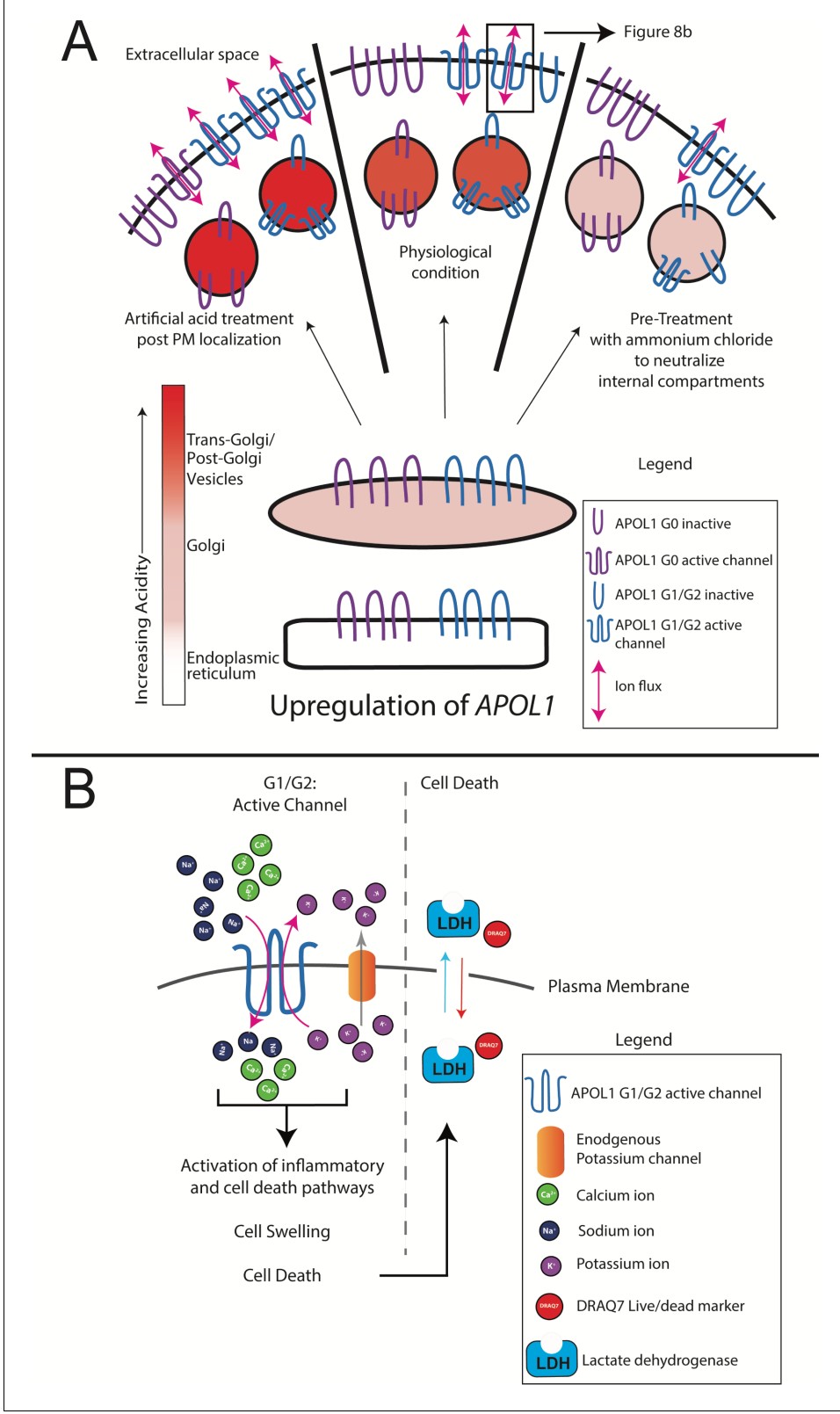

**Figure 8.** Model of RRV-mediated cytotoxicity: G1 and G2 form cation channels at the PM. (a) Proposed model of APOL1 trafficking and cytotoxicity. All variants of APOL1 will traffic to the PM, *en route* they will encounter acidification and neutralization along the secretory pathway, steps required for channel formation. However, while G1 and G2 are able to form cation channels when overexpressed, G0 does not. We hypothesize that G1 and G2 are more sensitive to pH-activation than G0, leading to channel formation. Artificial acidification of cells after localization of APOL1 to the PM

*Figure 8 continued on next page*

*Figure 8 continued*

caused G0 to become toxic. The increase in G1 and G2 cytotoxicity post-artificial acidification demonstrates that not all APOL1 at the PM is in a channel conformation. Protection against cytotoxicity due to pre-treatment with the weak base ammonium chloride signifies the requirement for acid-activation. (b) At the PM, G1 and G2 channels will lead to an influx of extracellular $Na^+$ and $Ca^{2+}$, initiating a cascade of events that eventually lead to cell death. Cell death is represented by the assays utilized in this study (release of cytoplasmic lactate dehydrogenase or influx of the live/dead stain DRAQ7).

was also exacerbated. Conversely, pre-treatment with ammonium chloride protected cells from RRV cytotoxicity. This demonstrates the importance of acidic activation for APOL1 channel activity, and that unlike G0, the RRVs more readily arrive at the PM in a channel active state.

It is imperative to elucidate the mechanism that prevents G0 cytotoxicity in order to understand how APOL1-mediated kidney disease manifests. Additionally, future work should determine how an acid-neutral pH gradient across a membrane activates APOL1 to form a channel, which may alter conformation or drive oligomerization. As can be seen in *Figure 7a*, G0 becomes cytotoxic only after additional acidification and neutralization. It may be that the RRVs are slightly more sensitive to acidification and therefore become activated more readily compared to G0, leading to kidney disease.

In summary, our results demonstrate that the kidney disease associated variants of APOL1 form cytotoxic cation channels at the cell surface. Live-cell analyses demonstrate an initial event leading to cell death is cation flux across the PM, with major ion components being extracellular $Na^+$ and $Ca^{2+}$. This ion flux precedes cell swelling by several hours and is therefore the likely driver of cell death. Because many of the reported pathways associated with APOL1 cell death and disease can be activated by pore-forming toxins and/or $Ca^{2+}$ signaling, we propose that the upstream event linking them is APOL1 channel activity at the PM. Taken together, our data strongly suggest that the primary focus for drug development should be prevention of G1 and G2 channel activity and targeting of activated APOL1 channels at the PM within the kidney.

# Materials and methods

**Key resources table**

| Reagent type (species) or resource | Designation | Source or reference | Identifiers | Additional information |
|---|---|---|---|---|
| Recombinant DNA reagent (*Homo sapiens*) | APOL1-G0 | NCBI | BC143038.1 | cDNA |
| Recombinant DNA reagent (*Homo sapiens*) | APOL1-G1 | NCBI | AF305428.1 | cDNA |
| Recombinant DNA reagent (*Homo sapiens*) | APOL1-G2 | 1000 genomes project, this paper | | cDNA *Constructed from mutagenesis from APOL1-G0. Protein coding sequence based off of 1000 genomes data |
| Recombinant DNA reagent PRG977 | PRG977 | Regeneron | | |
| Recombinant DNA reagent and transfected construct pcDNA5/FRT/TO | pcDNA5 | Thermo Fisher | V652020 | APOL1 variants cloned into this plasmid to generate stable cell line (FT293-APOL1_ |

*Continued on next page*

*Continued*

| Reagent type (species) or resource | Designation | Source or reference | Identifiers | Additional information |
|---|---|---|---|---|
| Recombinant DNA reagent and transfected construct pOG44 | p0G44 | Thermo Fisher | V600520 | |
| Recombinant DNA reagent and transfected pcDNA6/Tet-repressor | pcDNA6/Tet-repressor | Thermo Fisher | R25001 | |
| Recombinant DNA reagent and transfected Str-KDEL-SBP-EGFP-GPI | RUSH | Addgene | 65293 | Gift from Franck Perez. APOL1 variants cloned into this plasmid for transfection into cells (RUSH-APOL1). GFP and GPI anchor removed |
| Recombinant DNA reagent and transfected pGP-CMVB-GCaMP6f | GCaMP6f | Addgene | 40755 | A gift from Douglas Kim and the GENIE project |
| Recombinant DNA reagent and transfected CMV-ER-LAR-GECO1 | ER-LAR-GECO | Addgene | 61244 | A gift from Robert Campbell |
| Recombinant DNA reagent and transfected CMV-FliCR | FliCR | Addgene | 74142 | A gift from Robert Campbell |
| Sequence based reagent | APOL1_G0 K150E mutagenesis primers | This paper | PCR primer pair | F:5'TGAAAGAGTTTCCTCGGTTGAAAAGTGAGCTTGAGGATAAC R:5'GTTATCCTCAAGCTCACTTTTCAACCGAGGAAACTCTTTCA |
| Sequence based reagent | APOL1-G0 E150 Conversion to G1 mutagenesis Round 1 (S243G) | This paper | PCR primer pair | F:5'CGGATGTGGCCCCTGTAGGCTTCTTTCTTGTG R:5'CACAAGAAAGAAGCCTACAGGGGCCACATCCG |
| Sequence based reagent | APOL1-G0 E150 Conversion to G1 mutagenesis Round 2 (I384M) (Round 1 as template) | This paper | PCR primer pair | F:5'GGAGCTGGAGGAGAAGCTAAACATGCTCAACAATAATTATAAGA R:5'TCTTATAATTATTGTTGAGCATGTTTAGCTTCTCCTCCAGCTCC |
| Sequence based reagent | APOL1-G0 E150 Conversion to G12 mutagenesis | This paper | PCR primer pair | F: 5'AGCTAAACATTCTCAACAATAAGATTCTGCAGGCGGAC R: 5'GTCCGCCTGCAGAATCTTATTGTTGAGAATGTTTAGCT |

*Continued on next page*

*Continued*

| Reagent type (species) or resource | Designation | Source or reference | Identifiers | Additional information |
|---|---|---|---|---|
| Sequence based reagent | Insertion of APOL1 cDNA into pcDNA 5 vector | This paper | PCR primer pair | F: 5'ATGATATCGCCA CCATGGAGGGAGCTG R: 5'ATCTCGAGTCATCA CAGTTCTTGGTCCGCCTG |
| Sequence based reagent | Insertion of APOL1 cDNA into RUSH vector | This paper | PCR primer pair | F: 5'ATGCCCTGCAGGA GAGGAAGCTGG AGCGAGG R: 5'ATGCTCTAGA CTATCACAGTT CTTGGTCCGCC |
| Cell line (*Homo sapiens*) | HEK293 | ATCC | CRL-1573 | |
| Cell line (*Homo sapiens*) | FlpIn HEK 293 | Thermo Fisher | | Gift from Dr. Christian Brix Folsted Andersen. Converted into FlpIn TREX293 |
| Cell line (*Homo sapiens*) | Conditionally Immortalized Human podocytes | *Saleem et al., 2002* | | Gift from Dr. Moin Saleem and Dr. Jeffrey Kopp |
| Cell line (*Cricetulus griseus*) | CHO | ATCC | CCL-61 | |
| Antibody | Mouse anti-APOL1 | Proteintech | 66124–1-Ig | WB 1:2000 IF 1:800 |
| Antibody | Rabbit anti-APOL1 | Proteintech | 11486–2-AP | WB 1:5000 |
| Antibody | Rabbit anti-GAPDH | Proteintech | 10494–1-AP | WB 1:5000 |
| Antibody | Rabbit anti-Calnexin | Stressgen | SPA-860 | IF 1:200 |
| Antibody | Goat anti-mouse 680RD | LICOR | 92568070 | WB 1:10,000 |
| Antibody | Donkey anti-rabbit 800CW | LICOR | 925–32213 | WB 1:10,000 |
| Antibody | anti-rabbit Alexa 488 plus | Thermo Fisher | A32731 | IF 1:1500 |
| Antibody | anti-mouse Alexa 647 | Thermo Fisher | A21236 | IF 1:1000 |
| Chemical compound, drug | HCS Nuclear Mask | Thermo Fisher | H10325 | IF 1:400 |
| Chemical compound, drug | DRAQ7 | Abcam | ab109202 | Live cell microscopy 3 µM |
| Chemical compound, drug | Thapsigargin | Thermo Fisher | T7458 | |
| Chemical compound, drug | Interferon gamma | R and D Systems | 285IF100 | |

*Continued on next page*

*Continued*

| Reagent type (species) or resource | Designation | Source or reference | Identifiers | Additional information |
|---|---|---|---|---|
| Chemical compound, drug | Lactate dehydrogenase assay | Promega | G1781 | Cytotox 96 Non-Radioactive Cytotoxicity Assay |
| Commercial assay kit | MultiTox-Fluor Multiplex Cytotoxicity Assay | Promega | G9201 | |
| Commercial assay kit | Quik Change II Mutagenesis Kit | Agilent | 200523 | |
| Software | TrackMate | *Tinevez et al., 2017* | | |
| Software | Prism | GraphPad | | |
| Software | R-multicomp package | *Hothorn et al., 2008* | | |

## Cloning and vector construction

APOL1-G0 (BC143038.1) (Variant containing K150, *Figure 1a*) linear protein structure was determined using JPred (*Cole et al., 2008*). APOL1-G0 cDNA in the PRG977 plasmid was subjected to multiple rounds of mutagenesis using the QuikChange II Mutagenesis Kit (Agilent, Santa Clara, CA. 200523), to generate G1 (K150E, S342G, I384M) and G2 (K150E, NY del 388:389). APOL1 cDNA was inserted into the pcDNA5/FRT/TO (Thermo, Waltham, MA. V652020) and the Str-KDEL_SBP-EGFP-GPI (RUSH) (A gift from Franck Perez, Addgene 65293) (*Boncompain et al., 2012*) mammalian expression vectors. The APOL1 cDNA inserted into the RUSH contained the sequence just downstream of the signal peptide cleavage site at A27.

## Cell culture and transfections

FlpIn 293 cells from Thermo Scientific were first transfected with the pcDNA6/Tet-repressor plasmid and selected with Zeocin (Gibco, Waltham, MA. R25001) at 100 µg/mL and blasticidin (Gibco R21001) at 5 µg/mL. A single clone was expanded to generate all FT293-APOL1 cells. For single copy APOL1 cDNA cell lines, the Flp recombinase vector pOG44 (Thermo V600520) and APOL1 pcDNA5/FRT/TO were co-transfected at a 9:1 ratio. Selection was performed with blasticidin at 5 µg/mL and Hygromycin B (Thermo 10687010) at 150 µg/mL. Foci were pooled and polyclonal cell lines for APOL1 G0, G1, G2, and empty vector were expanded. Cells were then maintained in DMEM (Corning, Corning, NY. 10–017 CM) with 1 mM sodium pyruvate (Sigma, St. Louis, MO. P5280), 10% tet-free FBS, 100 µg/mL Hygromycin B, and 5 µg/mL blasticidin. All experiments were performed in the absence of antibiotics. APOL1 cDNA expression was induced in these cell lines with the addition of doxycycline (Sigma D9891) at 50 ng/mL unless otherwise stated.

HEK293 cells were cultured in DMEM + 10% FBS, CHO cells in F12K (ATCC 30–2004) + 10% FBS, and conditionally immortalized human podocytes (*Saleem et al., 2002*) in RPMI1640 (Corning 10–040-CV)+ 10% FBS and ITS (Gibco 41400045). Podocytes were maintained at 33C on Type-I collagen-coated plates. For experiments, podocytes were moved to 37C for 5–7 d to allow for differentiation, and then treated for 24 hr with the indicated amounts of Interferon-γ (R and D Systems, Minneapolis, MN. 285IF100). Cells were regularly tested for mycoplasma.

To perform $Na^+$ replacement and $Ca^{2+}$ reduction experiments, RUSH-APOL1 transfected HEK293 cells were cultured in Hank's Balanced Salt Solution supplemented with 10% fetal bovine serum, 4 mM L-glutamine, MEM amino acids (Thermo 11130051), MEM non-essential amino acids (Thermo 11140076), and 1 mM sodium pyruvate (approximately 20 mM $Na^+$ in total). For $Ca^{2+}$ reduction experiments, HBSS media was supplemented with 130 mM NaCl and $CaCl_2$ was serially diluted from 1.8 mM to 0.1125 mM. All plates were coated with 2.5 µg/cm$^2$ of fibronectin (Sigma F1141) to promote cell attachment in a low $Ca^{2+}$ environment. For $Na^+$ replacement, NaCl in media was reduced from 130 mM (150 mM total $Na^+$) to 65 mM and replaced with equal amounts of choline Cl or KCl. Cell death was assayed using the MultiTox-Fluor Multiplex Cytotoxicity Assay (Promega G9201) 12 hr after addition of biotin.

Cells were transfected with Lipofectamine 3000 as per the manufacturer's instructions. Cytotoxicity was measured via release of lactate dehydrogenase (LDH) using the Cytotox 96 Non-Radioactive Cytotoxicity Assay (Promega, Madison, WI. G1781). To simultaneously measure cytotoxicity and viability, the MultiTox-Fluor Multiplex Cytotoxicity Assay was used in combination with black-walled, optical bottom 96 well plates (Thermo 165305) and read on a Molecular Devices SpectraMax Gemini. Percent cell death was calculated first by determining the cytotoxicity/viability for each sample, followed by using minimum and maximum cell death controls.

## Lysate collection and immunoblotting

Lysates were collected in NP-40 lysis buffer (150 mM NaCl, 1.0 % NP-40, 1 mM EDTA, 50 mM Tris, pH 8.0) with HALT protease inhibitor (Thermo 78430). Total protein content was quantified with the DC protein assay (Bio-Rad, Hercules, CA. 5000112) and samples were diluted into 4x SDS Laemmli buffer with 2.5% β-mercaptoethanol, and equal amounts of protein were loaded into 10% Tris-Glycine SDS PAGE gels (Thermo XP00100). Blocking and antibody incubations were performed in Odyssey PBS Blocking Buffer (LICOR, Lincoln, NE. 927–40000) following the manufacturer's instructions. Primary antibodies used were mouse anti-APOL1 1:2000 (Proteintech, Rosemont, IL. 66124–1-Ig), rabbit anti-APOL1 1:5000 (Proteintech 11486–2-AP), and rabbit anti-GAPDH 1:5000 (Proteintech 10494–1-AP). Secondary antibodies used were goat anti-mouse 680RD 1:10,000 (LICOR 92568070), and donkey anti-rabbit 800CW 1:10,000 (LICOR 925–32213). Blots were scanned on a LICOR Odyssey Classic.

Electrophysiology rAPOL1 was purified from *E. coli* and analyzed in planar lipid bilayers as previously described (*Thomson and Finkelstein, 2015*). Briefly, planar lipid bilayers were formed from soybean asolectin, a rich phospholipid mixture from which non-polar lipids had been removed (*Kagawa and Racker, 1971*). In some experiments, cholesterol was added to increase bilayer stability. The lipid solution in pentane (1.0% asolectin w/v, or 1.5% asolectin, 0.5% cholesterol w/v) was layered on top of the aqueous solutions and the solvent allowed to evaporate. The lipid bilayer was formed by alternately raising the solution volumes above a ~ 100 micron hole in a Teflon septum separating symmetric 1 ml compartments as depicted in *Figures 2a* and *6a*; *Qiu et al., 1996*. The cis solution was defined as the side to which protein was added. The voltage was reported as that of the cis with respect to the trans. The trans-bilayer current due to APOL1 was measured in response to a voltage which was set by the experimenter. Manipulation of the pH was achieved by adding pre-calibrated volumes of HCl or KOH to the cis buffer.

To test for $Ca^{2+}$ permeability, bilayers were formed between identical solutions of 1) $CaCl_2$-buffer: 10 or 100 mM $CaCl_2$ (as detailed in *Figure 2* legend) 0.5 mM EDTA, 5 mM K-succinate, 5 mM K-HEPES, pH 7.1; or 2) excess KCl buffer: 150 mM KCl, 1 mM $CaCl_2$, 0.1 mM EDTA, 5 mM K-MES, 5 mM K-HEPES, pH 7.2. A pH-dependent conductance was obtained with the addition of APOL1 as detailed in *Figure 2* and then 1 M $CaCl_2$ was titrated into the to the cis compartment at pH 7.2. The reversal potential ($E_{rev}$) was determined before and after $CaCl_2$ addition by adjusting the voltage until the current read zero. To calculate permeability ratios of calcium versus potassium (pCa/pK) in the presence of excess KCl we used the following derivation of the Lewis equation as described by *Jatzke et al., 2002*:

$$\Delta E_{rev} = \frac{RT}{2F} \ln \left( 1 + \frac{P_{Ca}}{P_k} \frac{4[Ca^{2+}]}{[K^+]} \right)$$

where $\Delta E_{rev}$ is the change in $E_{rev}$ due to a given change in $CaCl_2$ concentration and R, T and F have their usual meanings.

To determine relative permeabilities of monovalent cations $K^+$, $Na^+$ and $choline^+$, a pH-dependent conductance was achieved with APOL1 in excess KCl buffer and then the cis buffer was perfused sequentially with similar solutions in which the KCl was replaced with NaCl and then choline chloride. $E_{rev}$ (the bi-ionic potential) was determined with KCl still present on the trans side. Permeability ratios (pX/pK) were determined by substituting into the Goldman-Hodgkin-Katz equation.

## Live-cell microscopy

All live cell experiments were performed using Fluorobrite DMEM (Gibco A18967-01). FT293 cells were seeded onto black-walled, optical bottom 96 well plates that were freshly coated with 2.5 μg/

cm$^2$ of fibronectin. Cells were transfected with pGP-CMVB-GCaMP6f (A gift from Douglas Kim and the GENIE project, Addgene 40755) (*Chen et al., 2013*), and the next day were treated with or without 50 ng/mL doxycycline to induce *APOL1* expression along with the addition of 3 µM DRAQ7 (Abcam, Cambridge, United Kingdom. ab109202). Cells were imaged via widefield every 10 min at 20x.

For microscopy of RUSH-transfected cells, HEK293 and CHO-K1 cells were seeded onto fibronectin coated glass bottom 96-well (Grenier, Kremsmünster, Austria. 655892) or 24-well (Cellvis, Mountain View, CA. P24-1.5H-N) plates. Cells were co-transfected with APOL1 RUSH vectors and one or a combination of the following Ca$^{2+}$ sensors: GCaMP6f, CMV-ER-LAR-GECO1 (A gift from Robert Campbell, Addgene 61244) (*Wu et al., 2014*), or plasma membrane voltage sensor CMV-FliCR (A gift from Robert Campbell, Addgene 74142) (*Abdelfattah et al., 2016*). The next day, 80 µM biotin (Sigma B4639) was added to respective wells and cells were then imaged every 5 min at 10 or 20x. Sensor validations for GCaMP6f and ER-LAR-GECO were performed in FT293 cells using the SERCA pump inhibitor thapsigargin (Thermo T7458).

## Immunofluorescence

CHO cells were seeded onto glass bottom, 96 well plates and transfected with RUSH-APOL1 plasmids. 24 hr after transfection, cells were treated with or without 80 µM biotin every 30 min for 0–120 min. For cell surface immunostaining, cells were moved onto ice and blocked with HBSS + Ca$^{2+}$ + Mg$^{2+}$ + 0.5% BSA fraction V, stained with primary antibodies, fixed in 2% formaldehyde (Thermo 28906), quenched with 50 mM NH$_4$Cl, and then stained with secondary antibodies. For intracellular staining, cells were permeabilized with 0.075% saponin. Staining was performed with the following antibodies and dyes: mouse anti-APOL1 1:800, rabbit anti-calnexin 1:200 (Stressgen, Farmingdale, NY. SPA-860), anti-rabbit Alexa 488 plus 1:1500 (Thermo A32731), anti-mouse Alexa 647 1:1000 (Thermo A21236), and HCS Nuclear Mask 1:400 (Thermo H10325). Cells were imaged via spinning disk confocal microscopy.

## Microscopy analysis

To measure the Ca$^{2+}$ kinetics in individual cells, fields of view (FOVs) were imaged at random and the corresponding video files were imported into Fiji and analyzed with TrackMate (*Tinevez et al., 2017*). After automated detection, files were manually curated to remove cells that were dead, overlapping, or those that had migrated out of the FOV. Cells were tracked based upon GCaMP6f signal, and the multi-channel tracking plug-in (https://imagej.net/TrackMate#Extensions) was used to collect data from all other channels within the spot (DRAQ7, ER-LAR-GECO, FliCR). The raw data was then exported and analyzed using R to determine the change in mean fluorescence intensity for each cell using the following equation:

$$(\text{Fluorescence}_{\text{Time=X}} - \text{Fluorescence}_{\text{Time=0}})/\text{Fluorescence}_{\text{Time=0}} = \Delta F/F0$$

where F$_0$ is the average mean fluorescence of the first 3 timepoints for each cell. For analysis with ER-LAR-GECO or FliCR, cells co-expressing GCaMP6f and exhibiting the established phenotype of Ca$^{2+}$ influx and cell swelling were analyzed.

To determine APOL1 trafficking kinetics in non-permeabilized immunostained cells, multiple FOVs were imaged at random at 20x. Image files were exported as maximum intensity Z-projections and analyzed using the scikit-image library in Python (*van der Walt et al., 2014*). Briefly, Nuclear-Mask stained cell nuclei were segmented and dilated to approximate cell boundaries. Cells were then filtered for cell death through co-localization with the segmented calnexin channel. Finally, APOL1 stain intensity was summed for all cells individually, and the criteria for positive staining of APOL1 was defined as a cell having greater summed APOL1 stained intensity than the maximum summed intensity of any cell within the untransfected image sets. Data from ≥10 FOVs per experimental group were averaged over three replicates and plotted to determine the intensity and percent of cells expressing APOL1 at the PM for each timepoint and genotype.

Preparation of representative images and movies was performed in Fiji. Representative immunofluorescence images are all maximum intensity Z-projections of approximately 10, 0.22 µm slices.

## Microscopes

For live-cell microscopy, a widefield setup was used. Live imaging was performed with a Zeiss (Oberkochen, Germany) Axio Observer with 470, 555, or 625 nm LED excitation along with Zeiss filter cubes 38 (green), 20 (red), and 50 (far-red). Recording was performed using a sCMOS with 6.5 µm pixels (Hamamatsu, Hamamatsu City, Japan. Flash4.0 v2). Live experiments were performed within an incubation chamber at 37C with 5% $CO_2$ and humidity. For immunofluorescence, a Zeiss Axio Observer was fitted with a Yokogawa (Musashino, Japan) CSU-X1 spinning disk head. Excitation was performed with a 405, 488, or 639 nm laser. Recording was performed using a back-thinned EMCCD camera with 16 µm pixels (Photometrics, Tucson, AZ. Evolve 512).

## Statistics

All graphing and statistical analyses were performed with Graph Pad Prism or the multicomp package in R (*Hothorn et al., 2008*). Statistical significance was tested using one or two-way ANOVAs with multiple comparisons tests. All data are represented as mean with 95% confidence interval unless otherwise noted, and all relevant p-values are depicted directly on each graph.

Data derived from quantified immunofluorescence were fit to a general linear model, and a simultaneous multiple comparison procedure between genotypes and timepoints was conducted in R version 3.5.1. P-values were corrected for multiple comparisons via the Benjamini-Hochberg correction (*Benjamini and Hochberg, 1995*), and significance was set to a minimum of $p < 0.05$.

## Acknowledgements

We would like to thank Dr. Christian Brix Folsted Andersen and Gitte Ratz from Aarhus University for providing the stable cell line system. We would also like to thank Anibelky Almanzar and Ji-Won Kang for helping to establish our stable cell lines. We would like to show our gratitude to Dr. Livia Bayer, Dr. Irina Catrina, and Dr. Diana Bratu of Hunter College for providing insight and expertise that greatly assisted our fluorescent microscopy approach. Additionally, we are grateful to Dr. Olaf Anderson of Weill Cornell Medicine and Dr. Mitchell Goldfarb of Hunter College for their expert advice and suggestions on electrophysiology, as well as Dr. Alan Finkelstein of Albert Einstein College of Medicine for both his guidance and loaning of equipment. We also thank the Weill Cornell Medicine's Visual Function Core Facility for its expert live imaging resources. Finally, we gratefully acknowledge our sources of funding the NIH and NSF, R01GM34107 and IOS-1249166 respectively.

## Additional information

### Funding

| Funder | Grant reference number | Author |
| --- | --- | --- |
| National Institute of General Medical Sciences | R01GM34107 | Enrique Javier Rodriguez-Boulan |
| National Science Foundation | IOS-1249166 | Jayne Raper |

The funders had no role in study design, data collection and interpretation, or the decision to submit the work for publication.

### Author contributions

Joseph A Giovinazzo, Conceptualization, Resources, Data curation, Formal analysis, Supervision, Funding acquisition, Investigation, Visualization, Methodology, Writing - original draft, Writing - review and editing; Russell P Thomson, Conceptualization, Formal analysis, Supervision, Investigation, Methodology, Writing - review and editing; Nailya Khalizova, Validation, Investigation; Patrick J Zager, Software, Formal analysis, Visualization; Nirav Malani, Software, Formal analysis; Enrique Rodriguez-Boulan, Conceptualization, Resources, Funding acquisition; Jayne Raper, Conceptualization, Resources, Supervision, Funding acquisition, Project administration, Writing - review and editing; Ryan Schreiner, Conceptualization, Resources, Data curation, Supervision, Validation, Investigation, Visualization, Methodology, Project administration, Writing - review and editing

## Author ORCIDs

Joseph A Giovinazzo (ID) https://orcid.org/0000-0003-2608-7143
Patrick J Zager (ID) https://orcid.org/0000-0002-3993-9686
Ryan Schreiner (ID) https://orcid.org/0000-0002-7457-6606

## Decision letter and Author response

Decision letter https://doi.org/10.7554/eLife.51185.sa1
Author response https://doi.org/10.7554/eLife.51185.sa2

## Additional files

### Supplementary files

• Transparent reporting form

### Data availability

All data generated or analysed during this study are included in the manuscript and supporting files. Source data files have been provided for all main figures in Dryad.

The following dataset was generated:

| Author(s) | Year | Dataset title | Dataset URL | Database and Identifier |
|---|---|---|---|---|
| Giovinazzo JA, Thomson RP, Khali-zova N, Zager PJ, Malani N, Rodri-guez-Boulan E, Ra-per J, Schreiner R | 2020 | Data from: Apolipoprotein L-1 renal risk variants form active channels at the plasma membrane driving cytotoxicity | https://dx.doi.org/10.5061/dryad.1ns1rn8r3 | Dryad Digital Repository , 10.5061/dryad.1ns1rn8r3 |

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
