## [Decision Letter]

**Acceptance summary:**

This article contributes important new information concerning the ion permeability of the Apolipoprotein L-1 channels and suggests that this is an important cause of the renal pathology that is associated with some Apolipoprotein L-1 variants.

**Decision letter after peer review:**

Thank you for submitting your article "Apolipoprotein L-1 renal risk variants form active channels at the plasma membrane driving cytotoxicity" for consideration by *eLife*. Your article has been reviewed by three peer reviewers, and the evaluation has been overseen by a Reviewing Editor and Olga Boudker as the Senior Editor. The following individual involved in review of your submission has agreed to reveal their identity: Suzie Scales (Reviewer #2).

The reviewers have discussed the reviews with one another and the Reviewing Editor has drafted this decision to help you prepare a revised submission.

Summary:

The reviews of your manuscript were very mixed, which appears to reflect substantial controversy in the field. I have therefore decided that the manuscript should be reconsidered after results of further experiments are available. One reviewer felt that your demonstration that the APOL1 must reach the cell surface was sufficiently important on its own, but the others did not. None of the reviewers was convinced that calcium influx was the direct cause, rather than a consequence or symptom, of cell death. It was suggested that increased cellular calcium may result from increased potassium efflux instead. One reviewer indeed pointed out that LDH release and calcium influx appeared to be simultaneous. Clarification of this issue is critical.

Essential revisions:

Several experiments were suggested in order to determine the relationship between calcium influx and potassium efflux. These were thought to be feasible within the 2-month period that is usually allowed by *eLife*:

Cell experiments:

1) Place the cells in high potassium media ("CKCM" of Olabisi et al., 2016) during biotin addition to determine if this prevents calcium influx and help clarify which cation is primarily responsible.

2) Experiments with low to zero extracellular calcium and with intracellular calcium chelators (BAPTA-AM) should be performed: currently, the lowest calcium concentration used was, according to one reviewer, higher than that in the Bowmans capsule.

3) Bilayer experiments: Studies with lower calcium concentrations, and to measure the permeability ratio between Ca^2+^ and K^+^ (PCa/PK) are required.

4) If it turns out that potassium efflux is indeed the major primary effect, it is not clear that the results would be rated as sufficiently novel, at least by 2 of the three reviewers.

5) One reviewer had reservations concerning the combination of protein and cell haplotype used; obviously this cannot be changed but the concerns must be mentioned.

6) Showing "typical" results without any statistics is no longer acceptable; using standard deviation for three measurements is not really acceptable either, strictly speaking. Ideally, present all three measurements, either all on the figure or, if this is too confusing, place them in the supplement.

Reviewer #1:

The paper makes a really strong statement such as that the APOL1 mediated cytotoxicity is calcium mediated.

The key problem with the presented work that it does not show that the increased extracellular calcium is responsible for the cell death using a dox inducible system.

The presented data shows a dose and genotype dependent cell death, which has been shown before (Pollak and Susztak labs etc). It also shows a dose and genotype dependent increase in calcium. When cells die intracellular calcium will rise, so it is essential to show that the increase in intracellular calcium actually a cause of the cell death not the consequence of the cell death. This is a fundamental flaw in the presented work.

It is not clear whether APOL1 actually conducts calcium it would be essential to demonstrate this claim. Using bilayer studies removing calcium and substituting calcium etc. are key for such statement.

Figure 2E the control and the experiments are presented on 2 separate graphs. They should be measured and presented together and statistical analysis need to be performed at 30 hours before and after injection. Simply put cells die over 30 hours we need to see the difference between APOL1 and control cells.

Same issue for Figure 3E.

The biotin is a clever experiment.

Another fundamental issue with the model is that it depends on putative and mysterious chaperone that has been identified. Otherwise the model would not work. There is ample of APOL1 in the circulation and could randomly insert to ant plasma membrane inducing cytotoxicity, however the disease and toxicity is high specific for the podocytes.

Reviewer #2:

Giovinazzo et al. demonstrate that renal risk variants (RRVs, G1 and G2) of APOL1, but not wild type (G0) in HEK293 and CHO cells lead to cytotoxicity via cell swelling and lysis, an important confirmation of results from several other labs because they use naturally occurring variants of APOL1 for the RRVs, whereas the majority of labs merely mutate G1 and G2 into the RefSeq, yielding sequences that are not present in the 1000 Genomes Project. They further perform a series of elegant experiments using the RUSH (an ER retention and synchronized release) system to clearly demonstrate that APOL1 must reach the cell surface to mediate these cytotoxic effects. This clearly rules out ER stress as the mechanism of RRV-mediated toxicity, which is important because APOL1 is mainly localized to the ER, and the plasma membrane is more accessible to certain classes of potential therapeutics. They also provide evidence that recombinant APOL1 can conduct calcium ions across lipid bilayers, and that in APOL1 RRV-expressing cells calcium influx (monitored with GCaMP) precedes cytotoxicity. They suggest that this ubiquitous cation second messenger could explain why such a variety of seemingly disparate mechanisms of APOL1 action have been previously proposed. The data are well presented and represent an impressive advance in the field as they strongly suggest that targeting the ion channel activity of plasma membrane APOL1 should be a viable therapeutic avenue for APOL1 nephropathies, although it is not entirely clear which cation is the relevant one. This paper definitely merits publication, and I only have a few minor suggestions for strengthening their conclusions.

1) While calcium influx is clearly an upstream event in RRV-mediated cytotoxicity, it is not completely certain that calcium influx is the event that should be therapeutically targeted. As they mentioned in the Discussion, although the calcium influx is demonstrated to be relatively soon (≤30 mins) after the arrival of APOL1 at the plasma membrane, it cannot be excluded that calcium influx from an unrelated calcium channel is secondary to efflux of e.g. K^+^ ions (which they previously showed can also be conducted by recombinant APOL1 in bilayers), which has also been shown to be an upstream event in APOL1-expressing 293 cells by Olabisi et al., 2016. While measuring the relative timing of K^+^ efflux in the RUSH system is surely outside the scope of their time and resources, it should be relatively easy to place the cells in high potassium media ("CKCM" of Olabisi et al.) during biotin addition to determine if this prevents calcium influx and help clarify which cation is primarily responsible.

2) On a related note, the 100mM calcium concentration at which they showed recombinant APOL1 conducted calcium ions in the bilayer assay is an extreme condition, up to 50x higher than the range at which cytotoxicity was observed in cells. To properly claim that APOL1 is permeable to calcium ions, they need to calculate the permeability ratio between Ca^2+^ and K^+^ (pCa/pK).

3) Can the authors comment on the low level (~20%) of cytotoxicity obtained by the RRVs in the stable FT293 cell lines? Is this simply because the stable cells were polyclonal and not all expressing at sufficient levels to cause toxicity, given that O'Toole et al., 2018, have previously demonstrated that killing is dose-dependent? Is there any IF or FACS data demonstrating what percentage of stable cells actually express APOL1? It is particularly important to show that G0 is expressed in a similar % of cells to at least one of the RRVs since an equal signal by western blotting could come from e.g. twice the percentage of cells expressing half the amount of APOL1, which would impact the overall cytoxicity results, since it is likely threshold-dependent on a per-cell basis.

4) The authors propose in Figure 5C that a secretory chaperone that binds to APOL1-G0 better than the RRVs might explain why only the RRVs lead to calcium influx in cells when all three conduct equally in planar lipid bilayers. If this were the case, one might expect APOL1 G0 to arrive faster at the cell surface, which it does not. I wonder if an alternative explanation could be that the "chaperone" is actually more of a G0-channel closing protein acting at the cell surface (after the low pH Golgi membrane insertion step). Couldn't this be easily tested by adding NH_4_Cl to neutralize the channel after arrival of APOL1 at the plasma membrane (instead of before biotin addition) and see if it no longer inhibits the RRVs?

5) Since cytotoxicity correlates with extracellular calcium levels (at least from 2-8mM), it would be helpful to mention in the Discussion that Bowman's capsule has around 1.4mM calcium (according to Hautmann and Oswald (J. Urology 1983 129 p433)), as it means their results could be physiologically relevant.

6) It would be nice to show the no biotin controls at equivalent time points in the LDH assay in Figure 3B rather than just stating there was no effect. P-value asterisks in this figure need to be larger.

Reviewer #3:

In the current manuscript the authors examine the hypothesis that APOL1 mediated cell injury and thereby kidney disease is mediated by plasma membrane calcium channel activity. The authors demonstrate that APOL1 expression in lipid bilayers leads to a calcium current. Calcium conductance was similar in APOL1 kidney disease non-risk (G0), and risk (G1 and G2) genetic variants. The authors used a combination of confocal immunofluorescence microscopy and the RUSH system for controlled and synchronous gene product expression and controlled compartment-specific trafficking in these studies. They demonstrate that APOL1 expression in the ER is innocuous to cell viability. Rather, intracellular calcium and cell injury measures, such as increased LDH release was associated with APOL1 trafficking and localization to the plasma membrane. Increased intracellular calcium preceded cell death (measured by the cell death marker DRAQ7). In spite of similar calcium conductance and plasma membrane expression of the non-risk and renal risk genetic variants (RRVs) only the RRVs led to increased intracellular calcium and cell death. In addition, lowering the pH of the medium, enhanced cell injury in RRVs (and also infor WT APOL1), confirming the previously reported importance of an acidic milieu for APOL1 channel activity. While the increased cellular calcium can explain the different perturbations in several cellular pathways mediated by APOL1, there remain a number of questions: Is APOL1 a selective calcium channel? Does the measured increase in intracellular calcium reflect a cell-injury causal calcium channel opening or rather a downstream reflection of another primary pathway of cell injury, wherein the RRVs are more injurious compared to G0 and hence also results in greater calcium influx? What is the role of alkaline pH, that was demonstrated as an essential factor for APOL1 cation conductivity in previous publications?

Following are some specific comments that are suggested for consideration by the authors:

1) The authors are referred to Lannon et al., 2019. This report demonstrates that the appropriate background haplotype for APOL1 G0 for comparison with G1 and G2, is that which naturally occurs in the population in which G1 and G2 occur, and not a construct on a haplotype which does not occur in natural populations at all (namely G1 and G2 mutagenesis on the reference European haplotype APOL1-G0 (BC143038.1), as described in the current manuscript). Thus for example, G0 K150E is itself was more toxic than G0 E150K. In the current manuscript the G0 variant used had K (lysine) at residue 150 whereas RRV had 150E (the natural variant) (Figure 1). G0 was innocuous to cells as opposed to RRV. Even though the frequency of this variant is higher than G0 with E, the G0K150E should also be used in the comparative cellular studies to examine the significance of glutamic acid at this position as an explanation for the different cells toxicity and cellular calcium level in RRV vs. GO.

2) The authors titrated APOL1 protein expression in FT293-G0 cells to similar levels found in interferon-stimulated human podocytes (Figure 1—figure supplement 1A) in order to achieve physiologic expression. They conclude that 0.2 ng/mL doxycycline and 10 ng/mL interferon-γ lead to similar levels of APOL1. However, it appears that APOL1 expression after Dox induction (0.2 ng/mL doxycycline) is lower than its expression after interferon (most evident in lane 3-from the right). In addition, for Figure 1 and Video 1, a higher concentrations of Dox are reported – therefore the expression of APOL1 variants is expected to be higher than the claim of achieving physiologic levels similar to those which accompany interferon induction in the podocyte platform.

3) Increased extracellular CaCl_2_ in cell culture media (from basal 2mM up to 8 Mm) was used to demonstrate the role of calcium influx and calcium content in cell injury in cells expressing the RRV compared to G0. The clinical relevance of elevated non-physiological extracellular calcium concentrations used in these experiments is not clear. In addition, the increased calcium influx could represent the consequence and not cause of non-specific cell injury, similar to previous reports regarding potassium depletion. In order to prove a causal role of extracellular calcium and consequent increased cytoplasmic calcium, experiments with low to zero extracellular calcium and with intracellular calcium chelators (BAPTA-AM) should be performed. In other words to prove the authors' claim that ion flux precedes cell swelling by several hours and is therefore the likely driver of cell death, can the authors state whether intracellular calcium buffering was used and if so, did it attenuate APOL1 cell injury?

4) The APOL1 plasmid used in the RUSH platform did not contain its natural signal peptide (Figure 3A), but rather the IL-2 signal peptide was used. Why is that and were studies conducted with APOL1 harboring its own signal peptide?

5) It is clear that increased calcium in cytoplasm preceded cell death (measured by measured by cell death marker DRAQ7), however increased LDH may represent an even earlier marker for cell injury. Looking at Figure 3: the earliest data presented regarding LDH release after adding biotin was after 6 hours – at that time point RRV (especially G2) lead to cell injury measured by LDH release. GCaMP6f fluorescence (calcium indicator protein) started to increase 3 hours after adding biotin. As increased cellular calcium may represent a non-specific marker for cell injury – please provide parallel data for LDH release 3 hours after adding biotin, in order to understand if increased calcium in cytoplasm precedes LDH release. Moreover, looking at Figure 5, LDH release was increased 2 hours after adding biotin, simultaneously with the increased calcium in cytoplasm, suggesting that increased calcium influx represents a nonspecific marker for cell injury.

6) Figure 4B: the experiments were conducted in non-permeabilized CHO cells, however, the staining of APOL1 after adding biotin seems to be in the ER and not only in the plasma membrane. Where is the calnexin stain mentioned in the text (as a control for non-permeabilized cells)?

7) Lower levels of G2 were detected at the plasma membrane compared to G0 (Figure 4). The authors explain this by invoking increased G2 cyto-toxicity. However, it is stated that the calnexin signal was used to filter out permeabilized cells (Figure 4C), which should mitigate such an effect as the explanation. That decreased G2 plasma membrane expression on one hand, and increased toxicity on the other hand, does not reconcile with the hypothesis that APOL1 channel activity at the plasma membrane is the most proximate causal mediator for cell death.

8) Increased calcium in cytoplasm is expected to increase ER calcium (Figure 4, Wu et al., 2014). Calcium release from ER should not exceed uptake to the ER. However, the data presented in Figure 3—figure supplement 2, show stable concentration of calcium in the ER, without an expected increase.

9) The authors hypothesize that G0 is sequestered by an unknown chaperone while trafficking along the secretory pathway. In light of similar expression of all APOL1 variants in the plasma membrane, it seems that G0 sequestration does not occur prior to trafficking to the plasma membrane.

10) RUSH-APOL1 transfected cells were transiently acidified at pH 5.5 then returned to neutral pH. According to Figure 5A, acidification enhances cell injury for all APOL1 variants including G0. However, as reported previously and also in this current manuscript by electrophysiologic studies – APOL1 insertion into lipid bilayers requires an acidic melieu and has transient chloride permeability which turns into cationic conductance (potassium – as per Bruno et al., 2017) at neutral pH. Similarly, in the current manuscript (Figure 2A), there is a small transient current at pH 5.6, that is enhanced significantly at pH 7.1. In light of these findings one could expect that an acidic pH in essential for APOL1 toxicity. Moreover, changing into neutral pH (after priming with an acidic pH) is expected to increase APOL1 conductance and therefore cell injury mediated by this cationic channel conductance. The data presented in Figure 5, seem contradictory: return to neutral pH attenuates APOL1 cell injury. In addition, if indeed APOL1 RRVs form calcium channels at the plasma membrane which increase intracellular calcium concentrations in turn leading to cell death, then returning to neutral pH would be expected to lead to decreased calcium influx and correspondingly decreased intracellular calcium levels – the results of such experiment should be presented.

11) Since the authors did not demonstrate selectivity for calcium conductance caveats regarding author mechanisms need attention. For example, increased cellular calcium may result from increased potassium efflux and cellular potassium depletion that was reported in previous manuscripts (Olabisi et al., 2016), and as mentioned briefly by the authors in the Discussion. cell death and kidney disease should be further explored.

[Editors' note: further revisions were suggested prior to acceptance, as described below.]

Thank you for submitting your article "Apolipoprotein L-1 renal risk variants form active channels at the plasma membrane driving cytotoxicity" for consideration by *eLife*. Your article has been reviewed by two peer reviewers, and the evaluation has been overseen by a Reviewing Editor and Olga Boudker as the Senior Editor The following individuals involved in review of your submission have agreed to reveal their identity: Katalin Susztak (Reviewer #1); Suzie Scales (Reviewer #2).

I haven't asked the reviewers for more discussion since they are in broad agreement; but reviewer 2 (pasted below) has picked up some issues that should be fixed.

Giovinazzo et al. have addressed most of the major concerns, in particular showing that Na^+^ as well as Ca^2+^ influx is an upstream event leading to APOL1-RRV mediated toxicity, and that Ca^2+^ influx occurs at physiological concentrations. It is an important advance that influx of Na+/Ca^2+^, rather than efflux of K^+^ is the trigger for APOL1-mediated toxicity, as that will change the nature of potential screens for channel inhibitors. Furthermore, as previously mentioned, the knowledge that APOL1 has to arrive at the plasma membrane to mediate toxicity, and that cation channel activity is the function that needs inhibiting to limit progression of CKD is vital for future therapeutic efforts. I therefore recommend this manuscript for publication.

A couple of points that I think still need to be changed:

1) In the absence of immunofluorescence or FACS data for the polyclonal APOL1-293 cells, the statement that the cells are expressing at physiological levels (compared to endogenous APOL1 in podocytes +IFNγ) needs to be toned down. It is still possible that the 20% of cells that die are the 20% highest expressers, which are effectively diluted out by the remaining non-expressers and thus the actual cells with active channels are higher expressing than the podocytes, especially as cytotoxicity is dose-dependent.

2) I agree with reviewer #3 and Essential revisions point 5 that the wrong G0 control (K150 instead of E150) was used for these studies. It is irrelevant that non-Africans (with G0 K150) happen to outnumber Africans (with G0 E150) selected for sequencing in the 1000 genomes project. E150 G0 would thus have been the correct haplotype to use. However, it is also true that E150 would likely only marginally diminish the difference between G0 and G1/G2 and so is not worth redoing the whole dataset. Nonetheless, the authors need to acknowledge that E150 would have been a better control and refer to Lannon et al. for the fact that it would have only made a small difference to the results.

---

## [Author Response]

Essential revisions:Several experiments were suggested in order to determine the relationship between calcium influx and potassium efflux. These were thought to be feasible within the 2-month period that is usually allowed by eLife:Cell experiments:1) Place the cells in high potassium media ("CKCM" of Olabisi et al., 2016) during biotin addition to determine if this prevents calcium influx and help clarify which cation is primarily responsible.

To study the effects of high potassium media where we replace the Na^+^ with K^+^, as suggested, we first wanted to establish the effect of the new media on cell viability. Lowering Na^+^ from 150 mM to 52.5 mM, where the cation is replaced stoichiometrically with K^+^, led to a *40% decrease* in cell viability after 12 hours (Olabisi et al. assessed cytotoxicity/viability in their experiments after 24 hours in media with complete Na^+^ replacement to K^+^). We also find an effect, though less dramatic, 20% deceased viability, with a similar ion replacement to choline^+^. Viability was further reduced when Na^+^ was lowered to 20 mM with the 130mM balance being replaced with choline^+^ (see Figure 6—figure supplement 1A and B). Based on these findings, we avoided full replication of the CKCM conditions, and settled on lowering Na^+^ to 85 mM for further experimentation (see Figure 6C and E). An additional point is high extracellular K^+^ depolarizes the cell; addition of 50 mM KCl to standard media causes depolarization which is the original reason we chose not to address the Olabisi et al. publication and K^+^ in our initial manuscript submission (see Figure 6—figure supplement 2A). Ionic perturbations are very complicated in vivo and numerous factors need to be considered during interpretation of the data.

APOL1-G2 mediated Ca^2+^ influx is largely unaffected by partial replacement of Na^+^ with choline^+^ or K^+^ (Video 6). GCaMP6f and RUSH-G2 transfected cells were placed in media containing 150 mM Na^+^ or 85 mM Na^+^ replaced by 65 mM choline^+^/K^+^. The APOL1-G2 mediated Ca^2+^ influx was observed under all media conditions and occurred with similar timing. We conclude that the large Ca^2+^ influx occurs independent of K^+^ efflux.

2) Experiments with low to zero extracellular calcium and with intracellular calcium chelators (BAPTA-AM) should be performed: currently, the lowest calcium concentration used was, according to one reviewer, higher than that in the Bowmans capsule.

Extracellular Ca^2+^ was serially diluted in media from 1.8 mM to 0.1125 mM, and the cytotoxicity of the APOL1 renal risk variants (RRVs) were significantly reduced with Ca^2+^ levels ≤ 0.45 mM (Figure 6—figure supplement 1C and also Figure 6D). This suggests that Ca^2+^ participates in RRV-mediated cytotoxicity.

All experiments were performed with 1.8 mM CaCl_2_ unless otherwise noted (standard concentration in all variations of Eagle's / Essential Media). It was mentioned in the review that the Bowman's capsule contains approximately 1.4 mM CaCl_2._ While we have shown that lower Ca^2+^ levels inhibit RRV cytotoxicity, no significant reduction in cytotoxicity was detected with 0.9 mM CaCl_2_ (Figure 6—figure supplement 1C). Thus, in our assay we will unlikely see any effect by lowering CaCl_2_ from 1.8 to 1.4 mM.

Further, our attempts to chelate incoming intracellular Ca^2+^ with BAPTA-AM while in very low or no Ca^2+^ media failed. The cells easily detached once the Ca^2+^ drops below 0.1mM, making the experiment difficult to be performed. With 0.1mM Ca^2+^ in the media the trivial moles of BAPTA in the cell, relative to the bath moles, are easily saturated. This hurdle was something which would take a good deal of time to overcome and we believe the data we present are convincing enough to make the BAPTA experiment unnecessary. We clearly demonstrate reducing the extracellular Ca^2+^ reduces toxicity.

3) Bilayer experiments: Studies with lower calcium concentrations, and to measure the permeability ratio between Ca^2+^ and K^+^ (pCa/pK) are required.

Planar lipid bilayer experiments were performed to address the conductance of Ca^2+^ with physiological levels of K^+^ and Ca^2+^. We demonstrate that in the presence of CaCl_2_ alone, APOL1 channels are selective for Ca^2+^ over Cl^-^ (Figure 2B). For this experiment, symmetrical solutions of 150 mM KCl and 1 mM CaCl_2_ were used on both sides of the bilayer. Increasing cis CaCl_2_ to 2 mM led to a positive shift in the current and a negative shift in the reversal potential (Figure 2D). The pCa/pK permeability ratio was calculated as 0.6, demonstrating that the APOL1 channel conducts Ca^2+^ at physiological salt conditions. For comparison, the PNa/pK permeability ratio was calculated as 1.1 (Figure 6B). Additionally, the APOL1-G2 mediated cellular Ca^2+^ influx was unaffected by media containing high K^+^ (Video 6, see response to Essential revisions point .1).

4) If it turns out that potassium efflux is indeed the major primary effect, it is not clear that the results would be rated as sufficiently novel, at least by 2 of the three reviewers.

The high potassium media "CKCM" of Olabisi et al., kills the cells on its own as discussed in the response to Essential revisions point 1 (Figure 6—figure supplement 1A and B). We further show the partially rescue of cells from RRV cytotoxicity that can be attributed to high potassium media is not because of the high K^+^, but rather because of lower Na^+^ in the media (Figure 6C). This is clearly demonstrated, where Na^+^ replacement with the impermeant choline^+^ results in a similar level of rescue. Importantly, 65 mM choline^+^ did not have as strong of a negative effect on general cell viability as the 65 mM K^+^(Figure 6—figure supplement 1A).

As Na^+^ influx can cause a Ca^2+^ influx and cytoplasmic accumulation on its own, we attempted to discern when the RRV-mediated Ca^2+^ influx occurred relative to Na^+^. Due to a lack of suitable genetically encoded Na^+^ sensors, we opted to use the plasma membrane voltage sensor FliCR as a readout for Na^+^ influx in combination with the Ca^2+^ sensor GCaMP6f. Using live-cell microscopy, we found that the influx of Ca^2+^ occurred concurrently with the increase in membrane depolarization (Figure 6F, Figure 6—figure supplement 2 and 3, Video 5). This data, along with the conductance of Ca^2+^ in the presence of 150 mM K^+^ (see Figure 2D) suggest the APOL1-RRVs are directly conducting Ca^2+^ into the cell. The influx of Na^+^ likely also contributes as the Na^+^/Ca^2+^ exchanger will be disrupted. Additionally, as Na^+^ influx typically leads to cell swelling and depolarization, it will drive K^+^ efflux, which is a cellular response to both events.

The effect that lower Ca^2+^ and Na^+^ levels have in rescuing cells from RRV cytotoxicity, along with the additive rescue when combined (Figure 6D) and their simultaneous influx (Video 5 and related figures) suggest that intracellular accumulation of these two cations represent the primary and upstream effect of RRV mediated cell death and disease, rather than the specific efflux of K^+^.

5) One reviewer had reservations concerning the combination of protein and cell haplotype used; obviously this cannot be changed but the concerns must be mentioned.

We are of the belief that G0 K150 is the appropriate haplotype control for these experiments, as it is the most prevalent in the human population and is over an order of magnitude more prevalent than G0 E150. We are in agreement with reviewer 3 about the importance of using the correct haplotypes, which is why we constructed Figure 1A to give readers context of the common haplotype backgrounds and our rationale for using G0. We did, however, construct an FT293 stable cell line expressing G0 E150 and compared the cytotoxic effect with G1 (Author response image 1).

There was no detectable loss in cell viability, similar to G0 K150 (Author response image 2).

**Author response image 2. respfig2:** 

(The above two experiments were not performed together.)Due to these results, we did not see any benefit to using both G0 E150 or K150 for this study and decided to move forward with G0 K150 as our control.

6) Showing "typical" results without any statistics is no longer acceptable; using standard deviation for three measurements is not really acceptable either, strictly speaking. Ideally, present all three measurements, either all on the figure or, if this is too confusing, place them in the supplement.

Thank you for this comment. We apologize for our previous presentation of the data using representative figures. This has now been corrected, with p values shown directly on the graphs themselves and the N reported in the legends.

Reviewer #1:The paper makes a really strong statement such as that the APOL1 mediated cytotoxicity is calcium mediated.The key problem with the presented work that it does not show that the increased extracellular calcium is responsible for the cell death using a dox inducible system.The presented data shows a dose and genotype dependent cell death, which has been shown before (Pollak and Susztak labs etc). It also shows a dose and genotype dependent increase in calcium. When cells die intracellular calcium will rise, so it is essential to show that the increase in intracellular calcium actually a cause of the cell death not the consequence of the cell death. This is a fundamental flaw in the presented work.

We have demonstrated that the influx of Ca^2+^ occurs several hours prior to cell swelling and cell death (Figure 2, Figure 3 and clearly observable in all videos). These finding place aberrant Ca^2+^ accumulation as an upstream event relative to cell swelling and cell death, and not a product of death itself. Additionally, as mentioned in Essential revisions points 1 and 2 and depicted throughout Figure 5 and its supplements, we show that Ca^2+^ is one of the drivers of cell death. This is most clearly shown when extracellular Ca^2+^ is reduced, lowering G1 and G2 cytotoxicity (Figure 6D and Figure 6—figure supplement 1C).

It is not clear whether APOL1 actually conducts calcium it would be essential to demonstrate this claim. Using bilayer studies removing calcium and substituting calcium etc. are key for such statement.

We have clearly shown that APOL1 conducts Ca^2+^ and show it is selective for Ca^2+^ over Cl^-^ (Figures 2B-D). We have expanded this planar lipid bilayer experiment to be performed with more physiological conditions (150 mM K^+^, 1-2 mM Ca^2+^), as described in Essential revisions point 3 (Figures 2D). This data shows APOL1 retains its Ca^2+^ permeability in the presence of physiological salt conditions and that the APOL1 channel directly conducts Ca^2+^ at these physiological ion concentrations.

Figure 2E the control and the experiments are presented on 2 separate graphs. They should be measured and presented together and statistical analysis need to be performed at 30 hours before and after injection. Simply put cells die over 30 hours we need to see the difference between APOL1 and control cells.Same issue for Figure 3E.

We have presented the data with consideration of clarity to the reader. The important controls in what are now Figure 3B and Figure 4D are the “induced” G0 as compared to the “induced” RRVs, G1 and G2. The upper panel in the two data sets are additional controls we have included to be perfectly transparent that the systems are a bit leaky, data which we originally considered to be placed in the supplementary data. Upon consideration of your comment we are now placing the ANOVA P-values between induced and un-induced into a table in Figure 3—source data 1 and Figure 4—source data 1, for additional consideration for the readers.

The biotin is a clever experiment.

Thank you!

Another fundamental issue with the model is that it depends on putative and mysterious chaperone that has been identified. Otherwise the model would not work. There is ample of APOL1 in the circulation and could randomly insert to ant plasma membrane inducing cytotoxicity, however the disease and toxicity is high specific for the podocytes.

We accept this criticism and have removed the "chaperone" from our model. There is no direct evidence of a chaperone that affects cytotoxicity between the variants (VAMP8 was discovered to bind G0 more strongly than G1 or G2 by John Sedor's group, however it did not affect cell death). We now only mention the chaperone hypothesis in the

Discussion.

Regarding the reviewer’s concluding sentence, please see Reeves-Daniel et al., 2011, Lee et al., 2012 and Kozlitina et al., 2016. We find the kidney transplant data the most compelling, where RRV kidneys transplanted into patients have a shortened allograft survival. Conversely, recipient genotype has no effect on allograft survival.

Additionally, in the study by Merck, there was no correlation between circulating levels of APOL1 and kidney disease. Further we would like to draw attention to the evidence in Figure 1—figure supplement 1B, which demonstrates podocytes treated with 10 ng/mL interferon-γ have an induction of *APOL1*, as has been shown by others. Here we further demonstrate that at a comparable APOL1 protein level RRVs and not G0 APOL1 are toxic to cells Figure 1—figure supplement 1C. This is well in line with the clinical observation that prolonged interferon treatment is detrimental to homozygous RRV APOL1 patients, see Nichols et al., 2015 and Markowitz et al., 2010.

Reviewer #2:[…]1) While calcium influx is clearly an upstream event in RRV-mediated cytotoxicity, it is not completely certain that calcium influx is the event that should be therapeutically targeted. As they mentioned in the Discussion, although the calcium influx is demonstrated to be relatively soon (≤30 mins) after the arrival of APOL1 at the plasma membrane, it cannot be excluded that calcium influx from an unrelated calcium channel is secondary to efflux of e.g. K^+^ ions (which they previously showed can also be conducted by recombinant APOL1 in bilayers), which has also been shown to be an upstream event in APOL1-expressing 293 cells by Olabisi et al., 2016. While measuring the relative timing of K^+^ efflux in the RUSH system is surely outside the scope of their time and resources, it should be relatively easy to place the cells in high potassium media ("CKCM" of Olabisi et al.) during biotin addition to determine if this prevents calcium influx and help clarify which cation is primarily responsible.

We thank the reviewer for the generous comments. To respond, please first see our response to Essential revisions point 1. In summary, we have performed the suggested experiment and found that CKCM like conditions did not affect Ca^2+^ influx (Video 6), Additionally, Ca^2+^ influx occurs concurrently with Na^+^ influx (Figure 6, Figure 6—figure supplements 2 and 3, and Video 5), indicating that it is the earliest detectable event of channel activity alongside Na^+^.

We do not believe in specifically targeting the flux of one ion over another as a relevant treatment possibility, rather we suggest targeting the active-state APOL1 channel at the plasma membrane is a viable strategy.

2) On a related note, the 100mM calcium concentration at which they showed recombinant APOL1 conducted calcium ions in the bilayer assay is an extreme condition, up to 50x higher than the range at which cytotoxicity was observed in cells. To properly claim that APOL1 is permeable to calcium ions, they need to calculate the permeability ratio between Ca^2+^ and K^+^ (pCa/pK).

The initial selectivity experiment was performed at the higher than physiological concentrations to get a strong signal to allow for the best analysis. However, we fully accept the need to perform experiments in more physiological conditions and have done the suggested experiment.

The results are explained in our response to Essential revisions point 3. In summary, APOL1 conducts Ca^2+^ (at 1-2 mM) in the presence of 150 mM K^+^(Figures 2B and D) The pCa/pK was calculated to be 0.6. In comparison, the pNa/pK is 1.1 (Figure 6B).

3) Can the authors comment on the low level (~20%) of cytotoxicity obtained by the RRVs in the stable FT293 cell lines? Is this simply because the stable cells were polyclonal and not all expressing at sufficient levels to cause toxicity, given that O'Toole et al., 2018, have previously demonstrated that killing is dose-dependent? Is there any IF or FACS data demonstrating what percentage of stable cells actually express APOL1? It is particularly important to show that G0 is expressed in a similar % of cells to at least one of the RRVs since an equal signal by western blotting could come from e.g. twice the percentage of cells expressing half the amount of APOL1, which would impact the overall cytoxicity results, since it is likely threshold-dependent on a per-cell basis.

All the APOL1 expressing FT293 cell lines are polyclonal. They are derived from a single clone expressing the tet repressor, and then clones after transfection with APOL1 plasmid were pooled together (correct integration of the new plasmid leads to the loss of zeocin resistance and the gain of hygromycin b resistance). Unfortunately, we do not have single cell data of the FT293 cell lines to see if there is a difference in the number of cells expressing APOL1, and to what level. We find the varied expression to be typical even for clonal cell line. As with nearly all transgenes, promoters become differentially methylated and have variable expression on a cell to cell basis, even in clonal cell populations. This can be partially corrected by using a CpG free promoter or by implementing a “universal chromatin opening element” (UCOE) upstream of the promoter. Additionally, targeting the transgene into a safe-harbor locus such as the ROSA26-like region would have helped to keep the expression variability at a lower level, however this would have to be done in follow-up work. We appreciate this comment and will keep this in mind for future work.

As for the low level of cytotoxicity, if the experiment is extended higher levels of cell death will be observed. This could in part be due to expression of APOL1 from a single copy.

While we cannot provide more a satisfactory answer for the FT293 cells, the RUSH experiments are transient transfections and lead to similar protein levels between the variants. We can state that for the RUSH system, which gives similar results to the FT293 cells, that we expect a similar number of cells expressing G0. This is corroborated by the cell surface staining data in Figure 5D and E.

4) The authors propose in Figure 5C that a secretory chaperone that binds to APOL1-G0 better than the RRVs might explain why only the RRVs lead to calcium influx in cells when all three conduct equally in planar lipid bilayers. If this were the case, one might expect APOL1 G0 to arrive faster at the cell surface, which it does not. I wonder if an alternative explanation could be that the "chaperone" is actually more of a G0-channel closing protein acting at the cell surface (after the low pH Golgi membrane insertion step). Couldn't this be easily tested by adding NH_4_Cl to neutralize the channel after arrival of APOL1 at the plasma membrane (instead of before biotin addition) and see if it no longer inhibits the RRVs?

We appreciate the comment and have responded to the merit of the chaperone in our model in our last response to Reviewer #1. We have removed the chaperone from our model and now only mention the hypothesis briefly in the Discussion. As for potential function of a putative chaperone, we are mostly in agreement as to how it functions i.e. somewhere along the secretory pathway and somehow regulating or modulating channel activity. With the lack of any chaperone discovered to date, however, it certainly could be another mechanism to explain the discrepancy in cytotoxicity. We present experiments and observations that G1 and G2 more readily form channels at the acidic pH encountered along the secretory pathway.

As for the acid/neutral experiments, we apologize for not being clearer. Adding ammonium chloride prior to biotin addition allows us to increase the pH within the cell, limiting APOL1 activation in the biosynthetic pathway. Moving the pH towards neutral in the normally acidic Golgi apparatus, trans-Golgi network and endosomal compartments of the biosynthetic pathway is well-documented. In the experiment the APOL1 experiences less acidic pH as it transits the biosynthetic compartments. Our experiment with APOL1 in planar lipid bilayers suggests that APOL1 "activation" into a channel state is irreversible upon acidification. Then the only way to modulate its activity afterwards is to change the pH (neutral = open, acidic = closed). APOL1 arrives to the neutral environment of the plasma membrane after encountering an acidic environment along the way in the control setting.

5) Since cytotoxicity correlates with extracellular calcium levels (at least from 2-8mM), it would be helpful to mention in the Discussion that Bowman's capsule has around 1.4mM calcium (according to Hautmann and Oswald (J. Urology 1983 129 p433)), as it means their results could be physiologically relevant.

This comment was addressed in the Essential revisions point 2 response. We performed the recommended experiment of lowering extracellular Ca^2+^ rather than raising it to non-physiological levels. CaCl_2_ was serially diluted from 1.8 mM to 0.1125 mM, and a significant reduction in cell death was only observed at levels ≤ 0.45 mM. Therefore, we do not believe there would be a significant difference in cytotoxicity between 1.4 and 1.8 mM CaCl_2_.

6) It would be nice to show the no biotin controls at equivalent time points in the LDH assay in Figure 3B rather than just stating there was no effect. P-value asterisks in this figure need to be larger.

We apologize for the lack of clarity in our presentation of the data. The graph does contain the no biotin control, but it was not properly presented. The new Figure 4B is now more clearly labeled. As for the p-value asterisks, we have moved to reporting the p-values themselves on each graph.

Reviewer #3:[…] While the increased cellular calcium can explain the different perturbations in several cellular pathways mediated by APOL1, there remain a number of questions:Is APOL1 a selective calcium channel? Does the measured increase in intracellular calcium reflect a cell-injury causal calcium channel opening or rather a downstream reflection of another primary pathway of cell injury, wherein the RRVs are more injurious compared to G0 and hence also results in greater calcium influx?

We have demonstrated APOL1 to be a calcium channel, more specifically APOL1 is a cation channel. Selectivity of APOL1 for Ca^2+^ is further answered in detail in our responses to Essential revisions points 1 and 2. APOL1 is permeable to Ca^2+^ and can conduct Ca^2+^ in the presence of physiological K^+^ levels. As for a cell injury causing Ca^2+^ influx, we clearly show in all six videos that the Ca^2+^ influx occurs upstream of any cell swelling and the subsequent lysis. Our method for detecting lysis is an approximately 600 – 700 dalton cell impermeable dye, DRAQ7 which is fluorogenic when bound to DNA. If APOL1 formed large pores in the plasma membrane similar to streptolysin O, DRAQ7 would be immediately detectable. Additionally, we show in the new Figure 6 and related video and supplementary figures that Ca^2+^ influx occurs alongside a Na^+^ influx/mild depolarization and that they are the first measurable events of APOL1 channel activity.

What is the role of alkaline pH, that was demonstrated as an essential factor for APOL1 cation conductivity in previous publications?

Treatment of cells with ammonium chloride has been known to inhibit the lytic activity of human serum for decades (Raper et al., 1996). It is also well described that APOL1 requires two steps to function as a cation channel: an acidic pH activation step, driving an irreversible membrane association, and a neutral pH step to open the channel (Thomson and Finkelstein, 2015).

Bringing the pH to a less acidic value in the intracellular compartments of the cell can be accomplished with ammonium chloride. Ammonium chloride pre-treatment of trypanosomes and, we now show mammalian cells, before exposure to APOL1 inhibits APOL1 activation when the cell is exposed to APOL1. Our data shows APOL1 in the neutral ER is not cytotoxic. The transit of APOL1 through the biosynthetic pathway as it is delivered to the plasma membrane is where the APOL1 can be activated by acidity. This activation becomes fully mature only once the APOL1 is neutralized, which is the state of the plasma membrane. Our use of ammonium chloride is to bring up the pH of the normally acidic biosynthetic compartments. By reducing the acidity of the biosynthetic pathway before addition of biotin, we reduce the cytotoxicity of the RRVs of APOL1 in our RUSH based model of APOL1.

Following are some specific comments that are suggested for consideration by the authors:1) The authors are referred to Lannon et al., 2019. This report demonstrates that the appropriate background haplotype for APOL1 G0 for comparison with G1 and G2, is that which naturally occurs in the population in which G1 and G2 occur, and not a construct on a haplotype which does not occur in natural populations at all (namely G1 and G2 mutagenesis on the reference European haplotype APOL1-G0 (BC143038.1), as described in the current manuscript). Thus for example, G0 K150E is itself was more toxic than G0 E150K. In the current manuscript the G0 variant used had K (lysine) at residue 150 whereas RRV had 150E (the natural variant) (Figure 1). G0 was innocuous to cells as opposed to RRV. Even though the frequency of this variant is higher than G0 with E, the G0K150E should also be used in the comparative cellular studies to examine the significance of glutamic acid at this position as an explanation for the different cells toxicity and cellular calcium level in RRV vs. GO.

This comment is answered in detail in our response to Essential revisions point 5. We have made stable cells, which express G0 E150 and detected no cell death (see Author response images 1 and 2). Additionally, we are of the belief that the appropriate control is G0 K150, as it the most prevalent in the human population. It is not the scope of this study to compare variants of "wild-type" APOL1, but to understand why the variants are cytotoxic.

2) The authors titrated APOL1 protein expression in FT293-G0 cells to similar levels found in interferon-stimulated human podocytes (Figure 1—figure supplement 1A) in order to achieve physiologic expression. They conclude that 0.2 ng/mL doxycycline and 10 ng/mL interferon-γ lead to similar levels of APOL1. However, it appears that APOL1 expression after Dox induction (0.2 ng/mL doxycycline) is lower than its expression after interferon (most evident in lane 3-from the right). In addition, for Figure 1 and Video 1, a higher concentrations of Dox are reported – therefore the expression of APOL1 variants is expected to be higher than the claim of achieving physiologic levels similar to those which accompany interferon induction in the podocyte platform.

We only demonstrate the use physiological levels in Figure 1—figure supplement 1. This effort taken was to address comments made by O'Toole that physiological levels of RRV expression are not cytotoxic. However, in their studies they did not use any method to determine what was "physiological". We show that expressing similar amounts of APOL1 as a podocyte in culture does in fact lead to cell death. FT293-APOL1 expression being lower than the podocytes on the western blot further bolsters the argument that physiologically inducible APOL1 levels as occurs in patients being treated with interferons can be cytotoxic in podocytes.

The lower the levels of APOL1, the slower the time to death. In the 0.2 ng/mL dox induction experiments we report the level of cytotoxicity after 48 hours. We see a more rapid progression of cell death with higher levels of APOL1 induction, the condition we then use in the rest of the manuscript. It is worth noting that we and others have seen that extremely high level of APOL1 expression has a toxicity even with the G0 variants. We carefully engineered two model systems to hold the APOL1 expression levels to best represent the physiologically relevant biology.

We did not use 0.2 ng/mL dox for future experiments because it would be extremely difficult to perform long-term live cell imaging. At this low dose, RRV cytotoxicity is severely delayed, and imaging for 3-4 days would be near impossible due to cell motility, cell division, and phototoxicity.

3) Increased extracellular CaCl_2_ in cell culture media (from basal 2mM up to 8 Mm) was used to demonstrate the role of calcium influx and calcium content in cell injury in cells expressing the RRV compared to G0. The clinical relevance of elevated non-physiological extracellular calcium concentrations used in these experiments is not clear. In addition, the increased calcium influx could represent the consequence and not cause of non-specific cell injury, similar to previous reports regarding potassium depletion. In order to prove a causal role of extracellular calcium and consequent increased cytoplasmic calcium, experiments with low to zero extracellular calcium and with intracellular calcium chelators (BAPTA-AM) should be performed. In other words to prove the authors' claim that ion flux precedes cell swelling by several hours and is therefore the likely driver of cell death, can the authors state whether intracellular calcium buffering was used and if so, did it attenuate APOL1 cell injury?

Thank you for this comment, we have performed the suggested experiment and answered in detail in the manuscript and also in response to Essential revisions point 2. Lower extracellular Ca^2+^ leads to significantly lower RRV cytotoxicity at levels ≤ 0.45 mM Ca^2+^.

Further, the total Ca^2+^ molar chelation capacity of intracellular BAPTA is easily overwhelmed by a continuous flux of Ca^2+^ and cannot have a sustained protective effect for the cell. Continuous Ca^2+^ increases over several hours (see all videos) and perhaps some temporal toxicity effect could be parsed out of very carefully planned experiments, however we believe more simple demonstrations are presented and answer the concerns of the reviewers.

4) The APOL1 plasmid used in the RUSH platform did not contain its natural signal peptide (Figure 3A), but rather the IL-2 signal peptide was used. Why is that and were studies conducted with APOL1 harboring its own signal peptide?

All FT293 cells contained the native APOL1 signal peptide. The RUSH system contains the IL-2 signal peptide and has been used in over a hundred publications with various other proteins with no negative effect. The signal peptide is simply to get the protein translated in the ER. N-terminal signal sequences are well known to be cleaved subsequent to the docking of the ribosome to the ER and we can find no reason for concern as to which signal sequence is utilized.

5) It is clear that increased calcium in cytoplasm preceded cell death (measured by measured by cell death marker DRAQ7), however increased LDH may represent an even earlier marker for cell injury. Looking at Figure 3: the earliest data presented regarding LDH release after adding biotin was after 6 hours – at that time point RRV (especially G2) lead to cell injury measured by LDH release. GCaMP6f fluorescence (calcium indicator protein) started to increase 3 hours after adding biotin. As increased cellular calcium may represent a non-specific marker for cell injury – please provide parallel data for LDH release 3 hours after adding biotin, in order to understand if increased calcium in cytoplasm precedes LDH release. Moreover, looking at Figure 5, LDH release was increased 2 hours after adding biotin, simultaneously with the increased calcium in cytoplasm, suggesting that increased calcium influx represents a nonspecific marker for cell injury.

LDH is a cytoplasmic protein encompassing 4 subunits. The full-size protein is approximately 144 kDa. In Video 1 and Video 2, all cells are treated with a cell impermeable dye, DRAQ7, which is approximately 600 – 700 Da. LDH would only be released, at its earliest, when DRAQ7 is first detected. DRAQ7 is not detected until after cell lysis (see Video 2). Therefore, the Ca^2+^ influx occurs several hours prior to LDH release.

Regarding Figure 5 (now Figure 6) we apologize for the confusion and have corrected this. LDH was measured 12 – 24 h AFTER biotin release. The 2h signified that cells were acidified 2 h after biotin.

We find that Ca^2+^ influx is not a non-specific, downstream event that occurs after cell injury. Ca^2+^ influx is concurrent with a Na^+^ influx, which we find as the first measurable effects of RRV channel activity (see Figure 6 and response to Essential revisions points 2, 3 and 4).

6) Figure 4B: the experiments were conducted in non-permeabilized CHO cells, however, the staining of APOL1 after adding biotin seems to be in the ER and not only in the plasma membrane. Where is the calnexin stain mentioned in the text (as a control for non-permeabilized cells)?

The merged images contain all three channels: Calnexin, APOL1, and nuclear stain. Author response image 3 shows the images of the calnexin channel alone, which shows no signal over background:

**Author response image 3. respfig3:** 

7) Lower levels of G2 were detected at the plasma membrane compared to G0 (Figure 4). The authors explain this by invoking increased G2 cyto-toxicity. However, it is stated that the calnexin signal was used to filter out permeabilized cells (Figure 4C), which should mitigate such an effect as the explanation. That decreased G2 plasma membrane expression on one hand, and increased toxicity on the other hand, does not reconcile with the hypothesis that APOL1 channel activity at the plasma membrane is the most proximate causal mediator for cell death.Lower RUSH-G2 levels were also detected via western blot (Figure 4—figure supplement 1A). While Figure 5 shows lower amounts of total G2 at the cell surface, the fold change, depicted in Figure 5D and E, shows that the increase in cell surface localization after biotin addition is similar to G0 and G1. Basically, G2 can do "more with less." That explanation is corroborated by the higher cytotoxicity of G2 compared to G1 seen in multiple experiments.

In the stable cells, however, similar levels of protein expression were found between all three variants, and the same results for cytotoxicity and Ca^2+^ influx are observed.

The timing of events should make it clear, that APOL1 gets to the cell surface only after biotin addition, and within a short amount of time only then is Ca^2+^ influx detected.

8) Increased calcium in cytoplasm is expected to increase ER calcium (Figure 4, Wu et al., 2014). Calcium release from ER should not exceed uptake to the ER. However, the data presented in Figure 3—figure supplement 2, show stable concentration of calcium in the ER, without an expected increase.

In Figure 4 of Wu et al., 2014, extracellular Ca^2+^ begins at 0 mM, and then a 20% increase in fluorescence (with the ER sensor) is detected when extracellular Ca^2+^ is increased to 2 mM. It is true that the ER acts as a Ca^2+^ sink and buffer for the rest of the cell, however this experiment where a massive change in extracellular Ca^2+^ occurs cannot be compared to a cation channel acting at the plasma membrane.

It is also important to note that in our Figure 3—figure supplement 2, the sensors have vastly different Kd for Ca^2+^ as they are working in the range of 100s of nM (cytoplasm) to 100s of µM (ER). The APOL1 mediated Ca^2+^ influx may lead to the ER taking up some of the Ca^2+^, however if that amount is only in the 10s or 100s of nM, then it may not be detected by the sensor. The ER-LAR-GECO sensor was primarily designed to measure release of Ca^2+^ from the ER.

9) The authors hypothesize that G0 is sequestered by an unknown chaperone while trafficking along the secretory pathway. In light of similar expression of all APOL1 variants in the plasma membrane, it seems that G0 sequestration does not occur prior to trafficking to the plasma membrane.

Thank you for this comment. As mentioned in response to reviewer 1 point 5 and reviewer 2 point 4, we have decided to omit the chaperone from our working model as there is no evidence in the literature. It is only a hypothesis, which we now solely mention in the Discussion.

10) RUSH-APOL1 transfected cells were transiently acidified at pH 5.5 then returned to neutral pH. According to Figure 5A, acidification enhances cell injury for all APOL1 variants including G0. However, as reported previously and also in this current manuscript by electrophysiologic studies – APOL1 insertion into lipid bilayers requires an acidic melieu and has transient chloride permeability which turns into cationic conductance (potassium – as per Bruno et al., 2017) at neutral pH. Similarly, in the current manuscript (Figure 2A), there is a small transient current at pH 5.6, that is enhanced significantly at pH 7.1. In light of these findings one could expect that an acidic pH in essential for APOL1 toxicity. Moreover, changing into neutral pH (after priming with an acidic pH) is expected to increase APOL1 conductance and therefore cell injury mediated by this cationic channel conductance. The data presented in Figure 5, seem contradictory: return to neutral pH attenuates APOL1 cell injury. In addition, if indeed APOL1 RRVs form calcium channels at the plasma membrane which increase intracellular calcium concentrations in turn leading to cell death, then returning to neutral pH would be expected to lead to decreased calcium influx and correspondingly decreased intracellular calcium levels – the results of such experiment should be presented.

Sorry for the misunderstanding. The ammonium chloride experiment only works if it is done *before* APOL1 is exposed to acidified environments. It is used to neutralize the acidic compartments in the cell/trypanosome before exposure to APOL1. Once APOL1 has become activated via acidification, ammonium chloride will have no effect. The APOL1 channel requires two steps to function: Acidification to form a channel active followed by a neutral pH for the channel open.

11) Since the authors did not demonstrate selectivity for calcium conductance caveats regarding author mechanisms need attention. For example, increased cellular calcium may result from increased potassium efflux and cellular potassium depletion that was reported in previous manuscripts (Olabisi et al., 2016), and as mentioned briefly by the authors in the Discussion. cell death and kidney disease should be further explored.

The selectivity of APOL1 for Ca^2+^ over Cl^-^ was clearly demonstrated. This has been answered in more detail in the response to Essential revisions points 1 and 3.

[Editors' note: further revisions were suggested prior to acceptance, as described below.]

[…] A couple of points that I think still need to be changed:1) In the absence of immunofluorescence or FACS data for the polyclonal APOL1-293 cells, the statement that the cells are expressing at physiological levels (compared to endogenous APOL1 in podocytes +IFNγ) needs to be toned down. It is still possible that the 20% of cells that die are the 20% highest expressers, which are effectively diluted out by the remaining non-expressers and thus the actual cells with active channels are higher expressing than the podocytes, especially as cytotoxicity is dose-dependent.

We have toned-down the degree of confidence in achieving physiological podocyte expression levels of APOL1. While we did make single-integrated stable cell lines using the Flp-In technology under a tet-promoter, we agree softening our statements to be more accurate is well advised.

2) I agree with reviewer #3 and Essential revisions point 5 that the wrong G0 control (K150 instead of E150) was used for these studies. It is irrelevant that non-Africans (with G0 K150) happen to outnumber Africans (with G0 E150) selected for sequencing in the 1000 genomes project. E150 G0 would thus have been the correct haplotype to use. However, it is also true that E150 would likely only marginally diminish the difference between G0 and G1/G2 and so is not worth redoing the whole dataset. Nonetheless, the authors need to acknowledge that E150 would have been a better control and refer to Lannon et al. for the fact that it would have only made a small difference to the results.

We address our use of the G0 K150 control as being suboptimal relative to the G0 E150 control in the Discussion.